# Forward $\chi^2$ Divergence Based Variational Importance Sampling

**Chengrui Li, Yule Wang, Weihan Li & Anqi Wu**
School of Computational Science & Engineering
Georgia Institute of Technology, Atlanta, GA 30305, USA
{cnlichengrui,yulewang,weihanli,anqiwu}@gatech.edu

## Abstract

Maximizing the marginal log-likelihood is a crucial aspect of learning latent variable models, and variational inference (VI) stands as the commonly adopted method. However, VI can encounter challenges in achieving a high marginal log-likelihood when dealing with complicated posterior distributions. In response to this limitation, we introduce a novel variational importance sampling (VIS) approach that directly estimates and maximizes the marginal log-likelihood. VIS leverages the optimal proposal distribution, achieved by minimizing the forward $\chi^2$ divergence, to enhance marginal log-likelihood estimation. We apply VIS to various popular latent variable models, including mixture models, variational auto-encoders, and partially observable generalized linear models. Results demonstrate that our approach consistently outperforms state-of-the-art baselines, in terms of both log-likelihood and model parameter estimation. Code: https://github.com/JerrySoybean/vis.

## 1 Introduction

Given the latent variables $\boldsymbol{z}$ and the observed variables $\boldsymbol{x}$, how to find the optimal parameter set $\theta$ that produces the maximum marginal likelihood $p(\boldsymbol{x};\theta) = \int p(\boldsymbol{x},\boldsymbol{z};\theta)\,\mathrm{d}\boldsymbol{z}$ is essential in a wide range of downstream applications. However, when the problem is complicated, we only know the explicit form of $p(\boldsymbol{x},\boldsymbol{z};\theta)$ and it is intractable to compute the marginal $p(\boldsymbol{x};\theta)$ analytically. Therefore, we turn to approximation methods such as variational inference (VI) (Blei et al., 2017) and importance sampling (IS) (Kloek & Van Dijk, 1978) to learn the model parameter $\theta$ and infer the intractable posterior $p(\boldsymbol{z}|\boldsymbol{x};\theta)$.

VI uses a variational distribution $q(\boldsymbol{z}|\boldsymbol{x};\phi)$ to approximate the posterior $p(\boldsymbol{z}|\boldsymbol{x};\theta)$ with the difference as their reverse KL divergence $\mathrm{KL}(q(\boldsymbol{z}|\boldsymbol{x};\phi)\|p(\boldsymbol{z}|\boldsymbol{x};\theta))$, where minimizing the KL divergence is equal to maximizing the evidence lower bound $\mathrm{ELBO}(\boldsymbol{x};\theta,\phi)$ of $\ln p(\boldsymbol{x};\theta)$. However, maximizing $\ln p(\boldsymbol{x};\theta)$ using ELBO may not be a good choice when dealing with complex posterior distributions, such as heavy-tailed or multi-modal distributions. There is chance that $\mathrm{KL}(q(\boldsymbol{z}|\boldsymbol{x};\phi)\|p(\boldsymbol{z}|\boldsymbol{x};\theta))$ is very small, but in fact both $q(\boldsymbol{z}|\boldsymbol{x};\phi)$ and $p(\boldsymbol{z}|\boldsymbol{x};\theta)$ are far from the true posterior $p(\boldsymbol{z}|\boldsymbol{x};\theta^{\mathrm{true}})$, leading to a higher ELBO but a lower marginal log-likelihood (e.g., Section 4.1).

Although other bounds such as $\alpha$ divergence-based lower bound (Li & Turner, 2016; Hernandez-Lobato et al., 2016) and $\chi^2$ divergence-based upper bound (Dieng et al., 2017) can be used for better posterior approximation, a more straightforward approach is to estimate $\ln p(\boldsymbol{x};\theta)$ by IS. Ideally, IS could have a good estimation if choosing a proper proposal distribution $q(\boldsymbol{z}|\boldsymbol{x};\phi)$ and a large number of Monte Carlo samples. In practice, however, there is often a lack of clear guidance on how to choose $q(\boldsymbol{z}|\boldsymbol{x};\phi)$ and limited indicators to verify the quality of $q(\boldsymbol{z}|\boldsymbol{x};\phi)$. Su & Chen (2021) showed that the variational distribution found by VI could serve as a proposal distribution for IS, but it is not the optimal choice (Jerfel et al., 2021; Saraswat, 2014; Sason & Verdú, 2016; Nishiyama & Sason, 2020). Besides, Pradier et al. (2019) noticed the numerical and scalability issue in minimizing forward $\chi^2$ divergence Finke & Thiery (2019), which should be treated rigorously.

To address these issues, we propose a novel learning method named variational importance sampling (VIS). We demonstrate that an optimal proposal distribution $q(\boldsymbol{z}|\boldsymbol{x};\phi)$ for IS can be achieved by minimizing the forward $\chi^2$ divergence in log space, which is numerically stable. Furthermore, with enough Monte Carlo samples, the estimated marginal log-likelihood $\ln\hat{p}(\boldsymbol{x};\theta)$ is an asymptotically tighter lower bound than ELBO, and hence $\ln p(\boldsymbol{x};\theta)$ could be maximized more effectively. In the experiment section, we apply VIS to several models including the most general case when there is no explicit decomposition $p(\boldsymbol{x},\boldsymbol{z};\theta) = p(\boldsymbol{x}|\boldsymbol{z};\theta)p(\boldsymbol{z};\theta)$, with both synthetic and real-world datasets

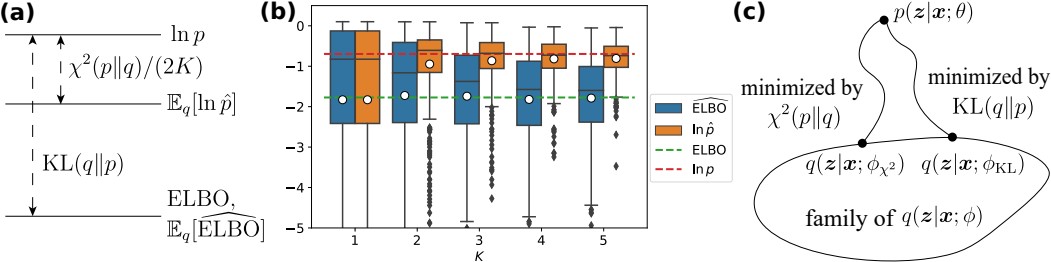

Figure 1: **(a)**: The bias between the marginal log-likelihood $\ln p(\boldsymbol{x}; \theta)$ and the expectation of its IS estimator $\mathbb{E}_q[\ln \hat{p}(\boldsymbol{x}; \theta, \phi)]$, the $\text{ELBO}(\boldsymbol{x}; \theta, \phi)$, and the expectation of the ELBO's estimator $\mathbb{E}_q[\widehat{\text{ELBO}}(\boldsymbol{x}; \theta, \phi)]$. When estimating $\ln p(\boldsymbol{x}; \theta)$, the down-biased IS estimator $\mathbb{E}_q[\ln \hat{p}(\boldsymbol{x}; \theta, \phi)]$ is a tighter lower bound than the down-biased ELBO estimator $\mathbb{E}_q[\text{ELBO}(\boldsymbol{x}; \theta, \phi)]$. **(b)**: Empirical visualization of the four quantities in (a) with different Monte Carlo samples $K \in \{1, 2, 3, 4, 5\}$. Each box in (b) is based on 500 repeats and the hollow circle on the box is their average. An asymptotic difference occurs when increasing $K$. **(c)**: Different $q(\boldsymbol{z}|\boldsymbol{x}; \phi)$ are obtained by minimizing the forward $\chi^2$ divergence, which is optimal for doing IS v.s. by minimizing the reverse KL divergence.

to demonstrate its superiority over the most widely used VI and three other state-of-the-art methods: CHIVI (Dieng et al., 2017), VBIS (Su & Chen, 2021), and IWAE (Burda et al., 2015). Appendix A.8 summarizes the related works and our corresponding contributions in a table.

## 2 BACKGROUND OF VARIATIONAL INFERENCE

Here we give a brief introduction to the variational inference (VI), its empirical estimator, and its bias. VI starts from the reverse KL divergence:

$$\text{KL}(q(\boldsymbol{z}|\boldsymbol{x}; \phi)\|p(\boldsymbol{z}|\boldsymbol{x}; \theta)) = \int q(\boldsymbol{z}|\boldsymbol{x}; \phi) \ln \frac{q(\boldsymbol{z}|\boldsymbol{x}; \phi)}{p(\boldsymbol{z}|\boldsymbol{x}; \theta)} \, \mathrm{d}\boldsymbol{z} = -\text{ELBO}(\boldsymbol{x}; \theta, \phi) + \ln p(\boldsymbol{x}; \theta), \quad (1)$$

with $\text{ELBO}(\boldsymbol{x}; \theta, \phi) := \mathbb{E}_q[\ln p(\boldsymbol{x}, \boldsymbol{z}; \theta) - \ln q(\boldsymbol{z}|\boldsymbol{x}; \phi)]$. Since ELBO is a lower bound of $\ln p(\boldsymbol{x}; \theta)$, the problem of maximizing $\ln p(\boldsymbol{x}; \theta)$ is converted to maximizing $\text{ELBO}(\boldsymbol{x}; \theta, \phi)$. VI is often favored for several reasons, such as: 1) The ELBO is formulated in terms of expectations of log-likelihood, making it numerically more stable compared to working directly with the original likelihood; 2) when the model can be factored as $p(\boldsymbol{x}, \boldsymbol{z}; \theta) = p(\boldsymbol{x}|\boldsymbol{z}; \theta)p(\boldsymbol{z}; \theta)$, the ELBO can be reformulated as $\text{ELBO}(\boldsymbol{x}; \theta, \phi) = \mathbb{E}_q[\ln p(\boldsymbol{x}|\boldsymbol{z}; \theta)] - \text{KL}(q(\boldsymbol{z}|\boldsymbol{x}; \phi)\|p(\boldsymbol{z}; \theta))$. This formulation is advantageous because the second KL term often has a closed-form expression for specific choices of the prior distribution $p(\boldsymbol{z}; \theta)$ and the variational distribution family $q(\boldsymbol{z}|\boldsymbol{x}; \phi)$, such as the Gaussian distribution.

In practice, the target function ELBO in Eq. 1 still requires numerical estimation, resulting in an empirical estimator

$$\widehat{\text{ELBO}}(\boldsymbol{x}; \theta, \phi) = \frac{1}{K} \sum_{k=1}^{K} \left[ \ln p\left(\boldsymbol{x}, \boldsymbol{z}^{(k)}; \theta\right) - \ln q\left(\boldsymbol{z}^{(k)} \middle| \boldsymbol{x}; \phi\right) \right], \quad (2)$$

where $\left\{\boldsymbol{z}^{(k)}\right\}_{k=1}^{K}$ are $K$ Monte Carlo samples from the variational distribution $q(\boldsymbol{z}|\boldsymbol{x}; \phi)$. Now, we convert maximizing $\ln p(\boldsymbol{x}; \theta)$ w.r.t. $\theta$ to maximizing $\widehat{\text{ELBO}}(\boldsymbol{x}; \theta, \phi)$ w.r.t. $\theta$ and $\phi$. The score function and pathwise gradient estimator of ELBO are shown in Appendix A.1.

**Bias of the ELBO estimator.** Note that although $\widehat{\text{ELBO}}(\boldsymbol{x}; \theta, \phi)$ is an unbiased estimator of ELBO, it is a strictly down-biased estimator of the marginal log-likelihood $\ln p(\boldsymbol{x}; \theta)$ (Fig. 1(a)):

$$\mathbb{E}_q\left[\widehat{\text{ELBO}}(\boldsymbol{x}; \theta, \phi) - \ln p(\boldsymbol{x}; \theta)\right] = \text{ELBO}(\boldsymbol{x}; \theta, \phi) - \ln p(\boldsymbol{x}; \theta) = -\text{KL}(q(\boldsymbol{z}|\boldsymbol{x}; \phi)\|p(\boldsymbol{z}|\boldsymbol{x}; \theta)). \quad (3)$$

As mentioned before, there is a chance that both $q(\boldsymbol{z}|\boldsymbol{x}; \phi)$ and $p(\boldsymbol{z}|\boldsymbol{x}; \theta)$ are far from the true posterior $p(\boldsymbol{z}|\boldsymbol{x}; \theta^{\text{true}})$, resulting in a higher ELBO but a lower marginal log-likelihood $\ln p(\boldsymbol{x}; \theta)$.

## 3 VARIATIONAL IMPORTANCE SAMPLING

To tackle this problem, we use importance sampling (IS) to estimate the marginal log-likelihood $\ln p(\boldsymbol{x}; \theta)$ directly. However, the approximation quality of IS depends on the choice of the pro-

posal distribution and the number of Monte Carlo samples. We first show that using IS can get an asymptotically tighter estimator of $\ln p(\boldsymbol{x}; \theta)$ than $\widehat{\mathrm{ELBO}}(\boldsymbol{x}; \theta, \phi)$. Then, we prove that the bias and effectiveness (variance) of this estimator are both related to the forward $\chi^2$ divergence and the number of Monte Carlo samples. This provides guidance on how to select the proposal distribution and the number of Monte Carlo samples. Finally, we derive the numerically stable gradient estimator used for obtaining the optimal proposal distribution.

**Down-biased IS estimator of the marginal log-likelihood.** With importance sampling (IS), the marginal is approximated via a proposal distribution $q(\boldsymbol{z}|\boldsymbol{x}; \phi)$, i.e.,

$$p(\boldsymbol{x}; \theta) = \int p(\boldsymbol{x}, \boldsymbol{z}; \theta) \, \mathrm{d}\boldsymbol{z} \approx \frac{1}{K} \sum_{k=1}^{K} \frac{p\left(\boldsymbol{x}, \boldsymbol{z}^{(k)}; \theta\right)}{q\left(\boldsymbol{z}^{(k)} \middle| \boldsymbol{x}; \phi\right)} =: \hat{p}(\boldsymbol{x}; \theta, \phi), \tag{4}$$

where $\left\{\boldsymbol{z}^{(k)}\right\}_{k=1}^{K}$ are $K$ Monte Carlo samples from the proposal distribution $q(\boldsymbol{z}|\boldsymbol{x}; \phi)$. For numerical stability, we need to work with it in log space,

$$\ln \hat{p}(\boldsymbol{x}; \theta, \phi) = \mathrm{logsumexp}\left[\ln p\left(\boldsymbol{x}, \boldsymbol{z}^{(k)}; \theta\right) - \ln q\left(\boldsymbol{z}^{(k)} \middle| \boldsymbol{x}; \phi\right)\right] - \ln K, \tag{5}$$

where the logsumexp trick can be utilized. Appendix A.2 shows that the gradient of $\ln p(\boldsymbol{x}; \theta)$ w.r.t. $\theta$ can be estimated as
$$\frac{\partial \ln p(\boldsymbol{x}; \theta)}{\partial \theta} \approx \frac{\partial \ln \hat{p}(\boldsymbol{x}; \theta, \phi)}{\partial \theta}. \tag{6}$$

Since
$$\mathbb{E}_q[\hat{p}(\boldsymbol{x}; \theta, \phi)] = \frac{1}{K} \sum_{k=1}^{K} \mathbb{E}_q\left[\frac{p(\boldsymbol{x}, \boldsymbol{z}; \theta)}{q(\boldsymbol{z}|\boldsymbol{x}; \phi)}\right] = \int p(\boldsymbol{x}, \boldsymbol{z}; \theta) \, \mathrm{d}\boldsymbol{z} = p(\boldsymbol{x}; \theta), \tag{7}$$

$\hat{p}(\boldsymbol{x}; \theta, \phi)$ is an unbiased estimator of $p(\boldsymbol{x}; \theta)$. However, $\ln(\cdot)$ is a concave function, thus $\mathbb{E}_q[\ln \hat{p}(\boldsymbol{x}; \theta, \phi)] \leqslant \ln \mathbb{E}_q[\hat{p}(\boldsymbol{x}; \theta, \phi)] = \ln p(\boldsymbol{x}; \theta)$ from Jensen's inequality. This means the estimator in log space $\ln \hat{p}(\boldsymbol{x}; \theta)$ is a down-biased estimator of $\ln p(\boldsymbol{x}; \theta)$.

**Bias of the IS estimator.** Similar to $\widehat{\mathrm{ELBO}}(\boldsymbol{x}; \theta, \phi)$, we can derive the bias of $\ln \hat{p}(\boldsymbol{x}; \theta, \phi)$ with the Delta method (Oehlert, 1992; Struski et al., 2022),

$$\mathbb{E}_q[\ln \hat{p}(\boldsymbol{x}; \theta, \phi) - \ln p(\boldsymbol{x}; \theta)] = \mathbb{E}_q\left[\ln\left(\frac{1}{K} \sum_{k=1}^{K} \frac{p\left(\boldsymbol{z}^{(k)} \middle| \boldsymbol{x}; \theta\right)}{q\left(\boldsymbol{z}^{(k)} \middle| \boldsymbol{x}; \phi\right)}\right)\right]$$

$$\approx -\frac{1}{2K} \mathrm{Var}_q\left[\frac{p(\boldsymbol{z}|\boldsymbol{x}; \theta)}{q(\boldsymbol{z}|\boldsymbol{x}; \phi)}\right] = -\frac{1}{2K}\left\{\mathbb{E}_q\left[\left(\frac{p(\boldsymbol{z}|\boldsymbol{x}; \theta)}{q(\boldsymbol{z}|\boldsymbol{x}; \phi)}\right)^2\right] - \mathbb{E}_q^2\left[\frac{p(\boldsymbol{z}|\boldsymbol{x}; \theta)}{q(\boldsymbol{z}|\boldsymbol{x}; \phi)}\right]\right\} \tag{8}$$

$$= -\frac{1}{2K}\left(\int \frac{p(\boldsymbol{z}|\boldsymbol{x}; \theta)^2}{q(\boldsymbol{z}|\boldsymbol{x}; \phi)} \, \mathrm{d}\boldsymbol{z} - 1\right) = -\frac{1}{2K}\chi^2(p(\boldsymbol{z}|\boldsymbol{x}; \theta) \| q(\boldsymbol{z}|\boldsymbol{x}; \phi)),$$

where $\chi^2(p\|q)$ is the forward $\chi^2$ divergence between $p$ and $q$ (Fig. 1(a)). Since Eq. 8 converges to 0 as $K \to \infty$, $\ln \hat{p}(\boldsymbol{x}; \theta, \phi)$ is an asymptotically tighter lower bound than $\widehat{\mathrm{ELBO}}(\boldsymbol{x}; \theta, \phi)$ (Fig. 1(a)). Particularly when $K = 1$, $\ln \hat{p}(\boldsymbol{x}; \theta, \phi) = \widehat{\mathrm{ELBO}}(\boldsymbol{x}; \theta, \phi)$. To verify this relationship empirically, we repeat the estimation of $\ln \hat{p}(\boldsymbol{x}; \theta, \phi)$ and $\widehat{\mathrm{ELBO}}(\boldsymbol{x}; \theta, \phi)$ based on $K$ Monte Carlo samples 500 times, and plot their empirical distributions w.r.t. $K$ in Fig. 1(b). With more Monte Carlo samples $K$, both $\ln \hat{p}$ and $\widehat{\mathrm{ELBO}}$ become stable, but the empirical expectation indicated by the hollow circle in each box implies that only $\ln \hat{p}(\boldsymbol{x}; \theta, \phi)$ converges to the marginal log-likelihood $\ln p(\boldsymbol{x}; \theta)$.

Fig. 1 demonstrates that IS can have a much better $\ln p(\boldsymbol{x}; \theta)$ estimation by setting a large $K$, which means using IS is a more direct way to maximize $\ln p(\boldsymbol{x}; \theta)$ than ELBO. Besides, to have a faster convergence, we also need to choose the proposal distribution $q(\boldsymbol{z}|\boldsymbol{x}; \phi)$ that minimizes $\chi^2(p(\boldsymbol{z}|\boldsymbol{x}; \theta) \| q(\boldsymbol{z}|\boldsymbol{x}; \phi))$ since this forward $\chi^2$ divergence could serve as an indicator of whether the proposal distribution is good: if the forward $\chi^2$ divergence is small, then the bias (the absolute value of Eq. 8) of the IS estimator is small.

On the other hand, we can write down the effectiveness (Freedman et al., 1998) of the estimator $\hat{p}(\boldsymbol{x}; \theta, \phi)$, i.e.,

$$\mathrm{Var}_q[\hat{p}(\boldsymbol{x}; \theta, \phi)] = \frac{1}{K^2} K \mathrm{Var}_q\left[\frac{p(\boldsymbol{z}|\boldsymbol{x}; \theta) p(\boldsymbol{x}; \theta)}{q(\boldsymbol{z}|\boldsymbol{x}; \phi)}\right] = \frac{p(\boldsymbol{x}; \theta)^2}{K} \chi^2(p(\boldsymbol{z}|\boldsymbol{x}; \theta) \| q(\boldsymbol{z}|\boldsymbol{x}; \phi)), \tag{9}$$

---

**Algorithm 1** VIS

---

1: **for** i = 1:N **do**
2:     Sample $\left\{ \boldsymbol{z}^{(k)} \right\}_{k=1}^{K}$ from $q(\boldsymbol{z}|\boldsymbol{x};\phi)$.
3:     Update $\theta$ by maximizing $\ln \hat{p}(\boldsymbol{x};\theta,\phi)$ via Eq. 6.
4:     Update $\phi$ by minimizing $\chi^2(p(\boldsymbol{z}|\boldsymbol{x};\theta)\|q(\boldsymbol{z}|\boldsymbol{x};\phi))$ via Eq. 12 or Eq. 24.
5: **end for**

---

which is the variance of the estimator. Eq. 8 and Eq. 9 coincide to indicate that for a small bias of $\ln \hat{p}(\boldsymbol{x};\theta,\phi)$ and a high effectiveness of $\hat{p}(\boldsymbol{x};\theta,\phi)$, we want a small $\chi^2(p(\boldsymbol{z}|\boldsymbol{x};\theta)\|q(\boldsymbol{z}|\boldsymbol{x};\phi))$ and a large $K$. In other words, we need as many Monte Carlo samples as possible; and the optimal choice of the proposal distribution for IS is the $q(\boldsymbol{z}|\boldsymbol{x};\phi)$ with the minimum forward $\chi^2$ divergence $\chi^2(p(\boldsymbol{z}|\boldsymbol{x};\theta)\|q(\boldsymbol{z}|\boldsymbol{x};\phi))$ rather than reverse KL divergence $\mathrm{KL}(q(\boldsymbol{z}|\boldsymbol{x};\phi)\|p(\boldsymbol{z}|\boldsymbol{x};\theta))$ (Fig. 1(c)).

The algorithm of the variational importance sampling (VIS) is summarized in Alg. 1. We first perform IS to maximize $\ln \hat{p}(\boldsymbol{x};\theta,\phi)$ w.r.t. $\theta$, given a fixed proposal distribution $q(\boldsymbol{z}|\boldsymbol{x};\phi)$; then we fix $\theta$ and minimize $\chi^2(p(\boldsymbol{z}|\boldsymbol{x};\theta)\|q(\boldsymbol{z}|\boldsymbol{x};\phi))$ w.r.t. $\phi$ to obtain a better proposal distribution for doing IS. However, minimizing $\chi^2(p(\boldsymbol{z}|\boldsymbol{x};\theta)\|q(\boldsymbol{z}|\boldsymbol{x};\phi))$ w.r.t. $\phi$ is non-trivial since we don't know $p(\boldsymbol{z}|\boldsymbol{x};\theta)$. We derive a stable gradient estimator for minimizing the forward $\chi^2$ divergence in the following.

**Gradient estimator.** Rewrite the forward $\chi^2$ divergence as

$$\chi^2(p(\boldsymbol{z}|\boldsymbol{x};\theta)\|q(\boldsymbol{z}|\boldsymbol{x};\phi)) = \frac{1}{p(\boldsymbol{x};\theta)^2} \int \frac{p(\boldsymbol{x},\boldsymbol{z};\theta)^2}{q(\boldsymbol{z}|\boldsymbol{x};\phi)} \, \mathrm{d}\boldsymbol{z} - 1 =: \frac{1}{p(\boldsymbol{x};\theta)^2} V(\boldsymbol{x};\theta,\phi) - 1. \quad (10)$$

So, minimizing $\chi^2(p(\boldsymbol{z}|\boldsymbol{x};\theta)\|q(\boldsymbol{z}|\boldsymbol{x};\phi))$ is equivalent to minimizing $V(\boldsymbol{x};\theta,\phi) := \int \frac{p(\boldsymbol{x},\boldsymbol{z};\theta)^2}{q(\boldsymbol{z}|\boldsymbol{x};\phi)} \, \mathrm{d}\boldsymbol{z}$ w.r.t. $\phi$. It still needs to be estimated and minimized in log space for numerical stability (Pradier et al., 2019; Finke & Thiery, 2019; Geffner & Domke, 2020; Yao et al., 2018). In Appendix A.3, we derive that $\ln V(\boldsymbol{x};\theta,\phi)$ can be estimated as

$$\ln V(\boldsymbol{x};\theta,\phi) \approx \mathrm{logsumexp} \left[ 2\ln p\left(\boldsymbol{x},\boldsymbol{z}^{(k)};\theta\right) - 2\ln q\left(\boldsymbol{z}^{(k)}\big|\boldsymbol{x};\phi\right) \right] - \ln K =: \ln \hat{V}(\boldsymbol{x};\theta,\phi).$$
$$(11)$$

The score function gradient estimator of $\ln V(\boldsymbol{x};\theta,\phi)$ w.r.t. $\phi$ at $\phi_0$ is

$$\frac{\partial \ln V(\boldsymbol{x};\theta,\phi)}{\partial \phi} \approx \frac{\partial}{\partial \phi} \frac{1}{2} \ln \hat{V}(\boldsymbol{x};\theta,\phi). \quad (12)$$

When the reparameterization trick can be utilized, $\boldsymbol{z}|\boldsymbol{x};\phi = g(\boldsymbol{\epsilon}|\boldsymbol{x};\phi)$ where $\boldsymbol{\epsilon} \sim r(\boldsymbol{\epsilon})$, then we have the transformation $q(\boldsymbol{z}|\boldsymbol{x};\phi) \, \mathrm{d}\boldsymbol{z} = r(\boldsymbol{\epsilon}) \, \mathrm{d}\boldsymbol{\epsilon}$ (Schulman et al., 2015). Now, we can get the pathwise gradient estimator $\frac{\partial \ln V(\boldsymbol{x};\theta,\phi)}{\partial \phi} \approx \frac{\partial}{\partial \phi} \ln \hat{V}(\boldsymbol{x};\theta,\phi)$, where we sample $\boldsymbol{\epsilon} \sim r(\boldsymbol{\epsilon})$ and use $\boldsymbol{z}^{(k)} = g\left(\boldsymbol{\epsilon}^{(k)}\big|\boldsymbol{x};\phi\right)$ in $\ln \hat{V}(\boldsymbol{x};\theta,\phi)$. The derivations are shown in Appendix A.3.

## 4 EXPERIMENTS

**Baselines for comparison.** We will apply VIS on three different models and compare it with four alternative methods:
- **VI**: The most widely used variational inference that maximizes ELBO.
- **CHIVI** (Dieng et al., 2017): When updating $\phi$, use both an upper bound CUBO (based on forward $\chi^2$ divergence) and a lower bound ELBO (based on reverse KL divergence) to squeeze the approximated posterior $q(\boldsymbol{z}|\boldsymbol{x};\phi)$ to the posterior $p(\boldsymbol{z}|\boldsymbol{x};\theta)$.
- **VBIS** (Su & Chen, 2021): Use the $q(\boldsymbol{z}|\boldsymbol{x};\phi)$ learned from VI as the proposal distribution of IS.
- **IWAE** (Burda et al., 2015): The importance-weighted autoencoder. It uses IS rather than VI to learn an autoencoder. An additional competitor for the VAE model only.

**Metrics.** For all models and datasets, we train the model with different methods on $\boldsymbol{x}_{\mathrm{train}}$ and evaluate on $\boldsymbol{x}_{\mathrm{test}}$ by: **marginal log-likelihood (LL)** $p(\boldsymbol{x}_{\mathrm{test}};\theta)$, which can be evaluated on both synthetic datasets and real-world datasets; **complete log-likelihood (CLL)** $p(\boldsymbol{x}_{\mathrm{test}},\boldsymbol{z}_{\mathrm{test}};\theta)$, which can be only evaluated on synthetic datasets, since we have the $\boldsymbol{z}_{\mathrm{test}}$ when generated the data; and **hidden log-likelihood (HLL)** $q(\boldsymbol{z}_{\mathrm{test}}|\boldsymbol{x}_{\mathrm{test}};\phi)$, which can be only evaluated on synthetic datasets for the same reason above.

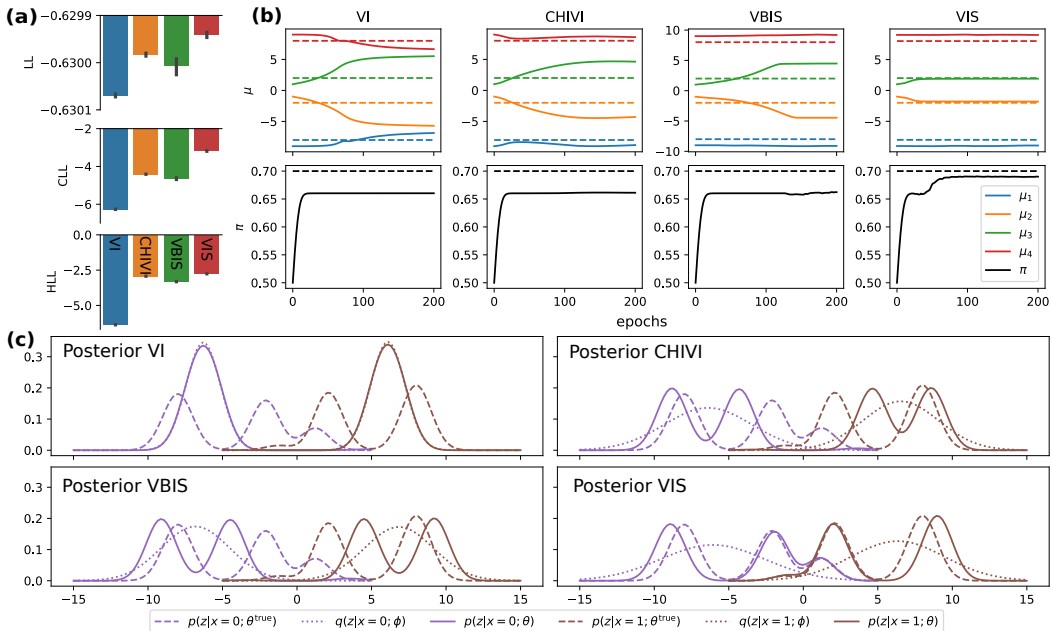

Figure 2: **(a)**: LL, CLL, and HLL evaluated on the test dataset. **(b)**: Convergence curves of the parameter set $\theta$ learned by different methods. The dashed curves are the true parameters used for generating the data, and the solid curves are the learned parameters. **(c)**: The posterior distribution given $x = 0$ and $x = 1$ in different methods. The dashed curves are the true posterior $p(z|x;\theta^{\mathrm{true}})$, the solid curves are the learned posterior $p(z|x;\theta)$, and the dotted curves are the approximated posterior $q(z|x;\phi)$ learned in the variational/proposal distribution.

## 4.1 A TOY MIXTURE MODEL

**Model.** We first use a toy mixture model to illustrate some representative behaviors of different methods. Consider the generative model $p(z;\theta) = \sum_{i=1}^{4} \pi_i \, \mathcal{N}(z;\mu_i, 1^2)$ with $\pi_1 = \pi_2 = \frac{1-\pi}{2}$, $\pi_3 = \pi_4 = \frac{\pi}{2}$; and $p(x|z;\theta) = \mathrm{Bern}(x; \mathrm{logistic}(z))$. The parameter set is $\theta = \{\pi\} \cup \{\mu_i\}_{i=1}^{4}$, the latent variable is $z \in \mathbb{R}$, and the observed variable is $x \in \{0,1\}$. Choosing the variational/proposal distribution family as $q(z|x;\phi) = \mathcal{N}(z;c_x,\sigma_x^2)$ for $x \in \{0,1\}$, and the variational/proposal parameter set is $\phi = \{c_0, c_1, \sigma_0, \sigma_1\}$. The simplicity of this model enables the visualization of $p(\boldsymbol{z}|\boldsymbol{x};\theta)$ for us to understand the behaviors of different methods.

**Experimental setup.** Both the training set and the test set consist of 1,000 samples simulated from the $p(x, z; \theta^{\mathrm{true}})$. We use Adam (Kingma & Ba, 2014) as the optimizer and the learning rate is set at 0.002. We run 200 epochs for each method, and in each epoch, 100 batches of size 10 are used for optimization. The number of Monte Carlo samples used for sampling the latent is $K = 5000$. We repeat 10 times with different random seeds for each method and report the performance.

**Results.** Quantitatively, VIS performs consistently better than all other methods in terms of all three metrics (Fig. 2(a)). In Fig. 2(b), we plot the convergence curves of the parameter set $\theta$ learned by different methods. Clearly, VIS achieves a more accurate parameter estimation. This further validates that a better parameter estimation corresponds to a higher test marginal log-likelihood.

To understand the effects of the approximated posterior $q(z|x;\phi)$ learned by different methods, we plot the true posterior $p(z|x;\theta^{\mathrm{true}})$ (dashed curves), the learned posterior $p(z|x;\theta)$ (solid curves), and the approximated posterior $q(z|x;\phi)$ (dotted curves) conditioned on $x = 0$ and $x = 1$ respectively in Fig. 2(c). First, we can tell that the true posterior $p(z|x;\theta^{\mathrm{true}})$ conditioned on both $x = 0$ and $x = 1$ are multi-modal shaped, with at least two distinct bumps. For example, $p(z|x = 0; \theta^{\mathrm{true}})$ has one large bump centered at about $z = -8$, one large bump centered at about $z = -2$, and one small bump centered at about $z = 1$ (see the purple dashed curve in Fig. 2(c)). Then we check the learned posterior $p(z|x = 0; \theta)$ and the approximated posterior $q(z|x = 0; \phi)$.

• For VI, the zero-forcing/mode-seeking behavior of minimizing the reverse KL divergence in VI makes the two large bumps on the left collapse into one. And the support of $q(z|x = 0; \phi)$ only

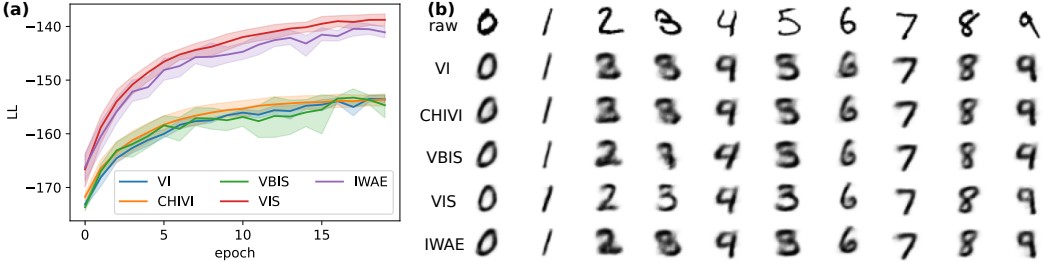

Figure 3: **(a)**: The marginal log-likelihood on the test set w.r.t. the training epoch. **(b)**: Examples of raw images and the reconstructed images by different methods.

covers the left large bump of $p(z|x = 0; \theta^{\text{true}})$, which leads to $p(z|x = 0; \theta)$ have very different shape to the $p(z|x = 0; \theta^{\text{true}})$. This is the case that the reverse KL divergence $\text{KL}(q(\boldsymbol{z}|\boldsymbol{x}; \phi)\|p(\boldsymbol{z}|\boldsymbol{x}; \theta))$ is very small, but in fact both $q(\boldsymbol{z}|\boldsymbol{x}; \phi)$ and $p(\boldsymbol{z}|\boldsymbol{x}; \theta)$ are far from the true posterior $p(\boldsymbol{z}|\boldsymbol{x}; \theta^{\text{true}})$, leading to a higher ELBO but a lower marginal log-likelihood.

- For VBIS, through importance sampling, the learned posterior $p(z|x = 0; \theta)$ maintains two large bumps but the small bump centered at about $z = 1$ is still not covered by $q(z|x = 0; \phi)$ due to the zero-forcing behavior of minimizing the reverse KL divergence. Besides, since the $q(z|x; \phi)$ learned by minimizing the reverse KL divergence is not the optimal proposal distribution for doing IS, the learned $p(z|x; \theta)$ is not good enough to match the true $p(z|x; \theta^{\text{true}})$ well.
- For CHIVI, both the reverse KL and the forward $\chi^2$ divergences are considered, so that the support of $q(z|x = 0; \phi)$ becomes much wider to make sure the density under both of the two large bumps can be sampled. However, it is still not wide enough to cover the small bump centered at about $z = 1$ compared with VIS. Besides, since CHIVI optimizes ELBO rather than the marginal log-likelihood w.r.t. $\theta$, the learned $\theta$ is not better than VIS.
- For VIS, the mass-covering/mean-seeking behavior of minimizing the forward $\chi^2$ divergence makes the $q(z|x = 0; \phi)$ wide enough to cover both the two large bumps and the small bump centered at about $z = 1$. Moreover, since we have shown in Eq. 8 and Eq. 9 that the $q(z|x; \phi)$ learned by minimizing the forward $\chi^2$ divergence is the optimal proposal distribution for doing IS, the shape of the learned posterior $p(z|x; \theta)$ matches the shape of the true posterior $p(z|x; \theta^{\text{true}})$ the best compared with other methods.

## 4.2 VARIATIONAL AUTO-ENCODER

**Model.** The generative model of a variational auto-encoder (VAE) (Kingma & Welling, 2013) can be expressed as $p(\boldsymbol{z}; \theta) = \mathcal{N}(\boldsymbol{z}; \mathbf{0}, \boldsymbol{I})$; and $p(\boldsymbol{x}|\boldsymbol{z}; \theta) = \text{Bern}(\boldsymbol{x}; \text{logistic}(\text{MLP}_{\text{dec}}(\boldsymbol{z})))$. The parameter set $\theta$ consists of all parameters of the MLP decoder. The variational/proposal distribution is parameterized as $q(\boldsymbol{z}|\boldsymbol{x}; \phi) = \mathcal{N}(\boldsymbol{x}; \boldsymbol{\mu}(\boldsymbol{x}), \boldsymbol{\sigma}^2(\boldsymbol{x})\boldsymbol{I})$, where $\boldsymbol{\mu}(\boldsymbol{x})$ and $\boldsymbol{\sigma}(\boldsymbol{x})$ are the output of the MLP encoder given input $\boldsymbol{x}$. The parameter set $\phi$ consists of all parameters of the MLP encoder.

**Experimental setup.** We apply the VAE model on the MMIST dataset (LeCun et al., 1998). There are 60,000 samples in the training set and 10,000 samples in the test set. Each sample is a $28 \times 28$ grayscale hand-written digit, so $\boldsymbol{x} \in [0, 1]^{784}$. For visualization, we set $\boldsymbol{z} \in \mathbb{R}^2$. Similar to (Kingma & Welling, 2013), we set the encoder and decoder structure as

$$\text{MLP}_{\text{dec}}(\boldsymbol{z}) = \boldsymbol{W}_{\text{dec},2}\boldsymbol{h}_{\text{dec}} + \boldsymbol{b}_{\text{dec},2}, \quad \boldsymbol{h}_{\text{dec}} = \tanh\left(\boldsymbol{W}_{\text{dec},1}\boldsymbol{z} + \boldsymbol{b}_{\text{dec},1}\right), \quad \boldsymbol{h}_{\text{dec}} \in \mathbb{R}^{128},$$

$$\begin{cases} \boldsymbol{\mu}(\boldsymbol{x}) = \boldsymbol{W}_{\boldsymbol{\mu}}\boldsymbol{h}_{\text{enc}} + \boldsymbol{b}_{\boldsymbol{\mu}} \\ \ln\boldsymbol{\sigma}(\boldsymbol{x}) = \boldsymbol{W}_{\boldsymbol{\sigma}}\boldsymbol{h}_{\text{enc}} + \boldsymbol{b}_{\boldsymbol{\sigma}} \end{cases}, \quad \boldsymbol{h}_{\text{enc}} = \tanh\left(\boldsymbol{W}_{\text{enc}}\boldsymbol{x} + \boldsymbol{b}_{\text{enc}}\right), \quad \boldsymbol{h}_{\text{enc}} \in \mathbb{R}^{128}. \tag{13}$$

We use Adam (Kingma & Ba, 2014) as the optimizer and the learning rate is set at $0.005$. We run 20 epochs for each method. The batch size is set as 64. The number of Monte Carlo samples used for sampling the latent is $K = 500$. We repeat 5 times with different random seeds for each method and report the test log-likelihood.

**Results.** Fig. 3(a) plots the marginal log-likelihood on the test set during learning. As the typical solver, VI performs roughly the same as CHIVI and VBIS, but the convergence curve of VI is a bit more stable. When comparing them with IWAE and VIS, however, IWAE is better and VIS is the best. The reconstruction images shown in Fig. 3(b) also imply that VAE solved by VIS could provide good reconstructions similar to the corresponding raw images. The learned latent manifolds by different methods are shown in Appendix A.4.

### 4.3 PARTIALLY OBSERVABLE GENERALIZED LINEAR MODELS

**Model.** We first present the classical generalized linear model (GLM) (Pillow et al., 2008), which studies multi-neuron interaction underlying neural spikes. We denote a spike train data as $Y \in \mathbb{N}^{T \times N}$ recorded from $N$ neurons across $T$ time bins, $y_{t,n}$ as the number of spikes generated by the $n$-th neuron in the $t$-th time bin. When provided with $Y$, a classic GLM predicts the firing rates $f_{t,n}$ of the $n$-th neuron at the time bin $t$ as

$$f_{t,n} = \sigma \left( b_n + \sum_{n'=1}^{N} w_{n \leftarrow n'} \cdot \left( \sum_{l=1}^{L} y_{t-l,n'} \psi_l \right) \right), \quad \text{with spike } y_{t,n} \sim \text{Poisson}(f_{t,n}), \quad (14)$$

where $\sigma(\cdot)$ is a non-linear function (e.g., Softplus); $b_n$ is the background intensity (bias) of the $n$-th neuron whose vector form is $\boldsymbol{b} \in \mathbb{R}^N$; $w_{n \leftarrow n'}$ is the weight of the influence from the $n'$-th neuron to the $n$-th neuron whose matrix form is $\boldsymbol{W} \in \mathbb{R}^{N \times N}$; $\boldsymbol{\psi} \in \mathbb{R}_+^L$ is the pre-defined basis function summarizing history spikes from $t - L$ to $t - 1$.

The classic GLM is not a latent variable model. However, we can extend a GLM to a partially observable GLM (POGLM) (Pillow & Latham, 2007), which becomes a latent variable model. Specifically, POGLM studies neural interaction when the spike data is partially observable, which is often the case in neuroscience since it is usually unrealistic to collect all neurons in a target brain region. Consider a group of $N$ neurons where $V$ of them are visible neurons and $H$ of them are hidden neurons (with $N = V + H$). Given the spike train $Y$, we denote its left $V$ columns as $X = Y_{1:T,1:V} \in \mathbb{N}^{T \times V}$ containing the visible spike train, and the right $H$ columns as $Z = Y_{1:T,V+1:N} \in \mathbb{N}^{T \times H}$ containing the hidden spike train. Then the firing rate is

$$f_{t,n} = \sigma \left( b_n + \sum_{n'=1}^{V} w_{n \leftarrow n'} \cdot \left( \sum_{l=1}^{L} x_{t-l,n'} \psi_l \right) + \sum_{n'=1+V}^{N} w_{n \leftarrow n'} \cdot \left( \sum_{l=1}^{L} z_{t-l,n'-V} \psi_l \right) \right), \quad (15)$$

for both visible and hidden neurons. Since the hidden spike train is not observable, POGLM becomes a latent variable model with observed variable $x_{t,n}$ and hidden variable $z_{t,n}$. The model parameter $\theta$ is defined to be $\{\boldsymbol{b}, \boldsymbol{W}\}$. The graphical model of POGLM is sketched in Fig. 4(a) top.

To do VI, VIS, or others on POGLM, a commonly used variational/proposal distribution (Rezende & Gerstner, 2014; Kajino, 2021) is $q(z_{t,n}|x_{1:t-1,1:V}, z_{1:t-1,1:H}) = \text{Poisson}(f_{t,n})$, where $f_{t,n}$ is defined in Eq. 15. Note that when using Eq. 15 to define the variational/proposal distribution, $\{\boldsymbol{b}, \boldsymbol{W}\}$ forms the variational/proposal parameter set $\phi$. The graphical model of the variational/proposal is sketched in Fig. 4(a) bottom.

#### 4.3.1 SYNTHETIC DATASET

**Experimental setup.** We randomly generate 10 different parameter sets $\theta$ of the GLM models for data generation, corresponding to 10 trials. There are $N = 5$ neurons in total, where the first $V = 3$ neurons are visible and the remaining $H = 2$ neurons are hidden. For each trial, we simulate 40 spike trains for training and 20 spike trains for testing. The length of each spike train is $T = 100$ time bins. The linear weights and biases of the model used for learning are all initialized as 0s. We use Adam (Kingma & Ba, 2014) as the optimizer and the learning rate is set at $0.01$. We run 20 epochs for each method, and in each epoch, 4 batches of size 10 are used for optimization. The number of Monte Carlo samples used for sampling the hidden spikes is $K = 2000$. We repeat 10 times with different random seeds for each method and report the performance.

**Results.** From the barplot in Fig. 4(b), we can see that VIS performs significantly better than the other three methods in terms of all three metrics (LL, CLL, and HLL). Similar to the toy mixture model, we can also check the parameter estimation and compare them with the true parameter set used for generating the data. The average weight and bias error are presented in the rightmost two bar plots in Fig. 4(b). The weight error of the VIS is the smallest. For the bias error, both VBIS and VIS are the smallest and are significantly smaller than VI and CHIVI.

In Fig. 4(c), we also visualize the parameter recovery results from different methods. For the bias vector, we can visually see that VI and CHIVI are worse than VBIS and VIS. For example, the bias of neuron 2 is positive, but only VIS recovers this positive value. For the visible-to-visible weights (the top-left block of the weight part), all four methods can match the true well. For the hidden-to-visible weights (the top-right block of the weight part), VI and CHIVI do not get enough gradient due to maximizing ELBO, so these weights are still kept around 0. For the visible-to-hidden

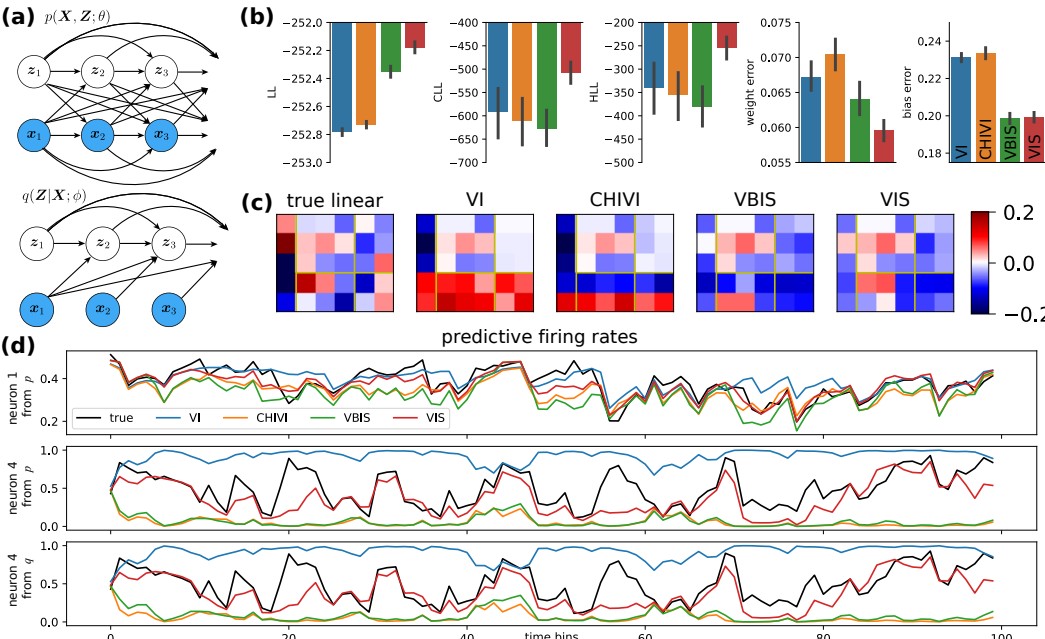

Figure 4: **(a)**: Graphical model of $p(\boldsymbol{X}, \boldsymbol{Z}; \theta)$ and $q(\boldsymbol{Z}|\boldsymbol{X}; \phi)$. **(b)**: The LL, CLL, HLL on the test set, and the average parameter error of the weights and biases in the linear mapping. **(c)**: True and estimated parameters by different methods of the first trial. For each matrix, the leftmost column is the bias $\boldsymbol{b}$, and the remaining block is the weight matrix $\boldsymbol{W}$. The top-left block of the weight part represents visible-to-visible, the top-right block represents hidden-to-visible, the bottom-left block represents visible-to-hidden, and the bottom-right block represents hidden-to-hidden. **(d)**: Predictive firing rates on a spike train from different methods. Specifically, given a complete test spike train $\boldsymbol{Y} = [\boldsymbol{X}, \boldsymbol{Z}]$, we can predict the firing rates by the complete model $p(\boldsymbol{X}, \boldsymbol{Z}; \theta)$ via Eq. 14 for both visible neurons (e.g., neuron 1) and hidden neurons (e.g., neuron 4). For hidden neurons (e.g., neuron 4), we can also predict the firing rates by $q(\boldsymbol{Z}|\boldsymbol{X}; \phi)$.

weights (the bottom-left block of the weight part), VI, CHIVI, and VBIS provide random-like and non-informative estimations, but VIS matches the true better. For the hidden-to-hidden weights (the bottom-right block of the weight part), none of the four methods gives acceptable results. The worse performances on the hidden-to-visible and hidden-to-hidden blocks also reflect the limitation of the variational/proposal distribution family.

In Fig. 4(d), we visualize the predictive firing rates $f_{t,n}$ learned by different methods. The top panel and the middle panel of Fig. 4(d) show that the firing rates predicted by $p(\boldsymbol{X}, \boldsymbol{Z}; \theta)$ obtained from VIS for both visible neurons and hidden neurons are the most accurate to the true firing rates among all four methods. Particularly, since only VIS learns acceptable visible-to-hidden weights, the firing rates predicted by VI, CHIVI, and VBIS are significantly worse than by VIS (the middle panel of Fig. 4(d)). These correspond to the CLL bar plot in Fig. 4(b). The bottom panel of Fig. 4(d) indicates that the proposal distribution of VIS can sample hidden spikes much closer to the true hidden spikes, which improves the learning effects and results in a better parameter recovery. Moreover, methods except VIS in the middle panel and the bottom panel of Fig. 4(d) reveal the case that $q(\boldsymbol{Z}|\boldsymbol{X}; \theta)$ and $p(\boldsymbol{Z}|\boldsymbol{X}; \theta)$ are close in terms of the reverse KL divergence, but both of them are far from the true posterior, resulting in higher ELBO but lower marginal log-likelihood than VIS.

### 4.3.2 RETINAL GANGLION CELL (RGC) DATASET

**Dataset.** We run different methods on a real neural spike train recorded from $V = 27$ retinal ganglion neurons while a mouse is performing a visual test for about 20 mins (Pillow & Scott, 2012). Neurons 1-16 are OFF cells, and neurons 17-27 are ON cells.

**Experimental setup.** We use the first $\frac{2}{3}$ segment as the training set and the remaining $\frac{1}{3}$ segment as the test set. The original spike train is converted to spike counts in every 50 ms time bins. For applying the stochastic gradient descent algorithm, we break the whole sequence into several pieces. The length of each piece is 100 time bins. First, we learn a fully observed GLM as a baseline. Then,

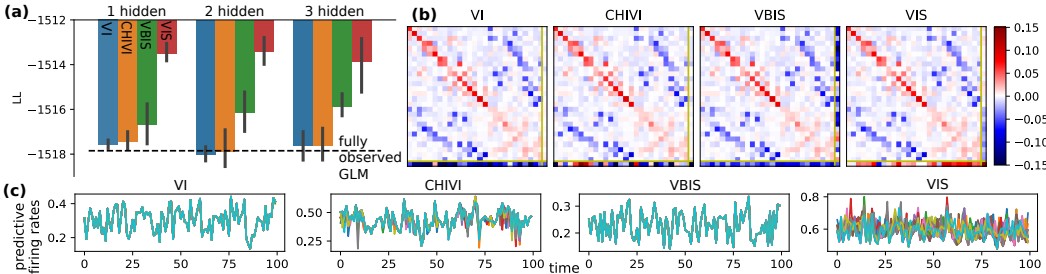

Figure 5: **(a)**: The marginal log-likelihood on the test segment with different numbers of hidden neurons. **(b)**: The estimated weight matrices from different methods. **(c)**: 20 predictive firing rates generated from 20 hidden spikes sampled from different variational/proposal distributions.

we assume there are $H \in \{1, 2, 3\}$ hidden representative neurons and learn the POGLM by different methods. We use Adam (Kingma & Ba, 2014) as the optimizer and the learning rate is set at $0.01$. We run 10 epochs for each method. The batch size is set as 32. The number of Monte Carlo samples used for sampling the hidden are 1,000, 2,000, and 3,000 for $H = 1, 2, 3$ respectively. We repeat 10 times with different random seeds for each method and report the performance.

**Results.** Compared with the fully observed GLM (the dashed line in Fig. 5(a)), adding hidden neurons significantly improves the capability of predicting spiking events on the test set, when learned by VBIS and VIS. This is reflected in the high test marginal log-likelihood of VBIS or VIS shown in Fig. 5(a). Particularly, VIS always obtains the highest test marginal log-likelihood compared with the three alternative methods.

We also visualize the learned weight matrix with one hidden neuron from the four methods in Fig. 5(b). With one hidden neuron learned by VIS, the weights from the hidden neuron to nearly all OFF cells are positive, and the weights to all ON cells are negative. This implies that this hidden representative neuron behaves like an OFF cell. The signs of the weights from this hidden representative neuron to the visible neurons clearly tell us the type of those visible post-synaptic neurons. All other methods do not have such a significant differentiation in the last column of the weight matrix.

Since we do not have the true hidden spike train in the real-world dataset, we sample hidden spike trains from the variational/proposal distribution $q(\boldsymbol{Z}|\boldsymbol{X}; \phi)$, and compute the corresponding firing rates that are used for sampling the hidden spike trains. In Fig. 5(c), we plot 20 randomly sampled predictive firing rates of the hidden neuron in the one-hidden-neuron model. Clearly, the predictive firing rates generated by VIS provide a wider effective support range for sampling, due to the mass-covering/mean-seeking behavior of minimizing the forward $\chi^2$ divergence. This variability improves the effectiveness of learning $\ln p(\boldsymbol{X}; \theta)$. Compared with VIS, the variational/proposal distributions learned by VI and VBIS are very restricted and concentrative, providing less variability in sampling hidden spikes. Since CHIVI minimizes both the forward $\chi^2$ and the reverse KL divergences, the variability of the variational/proposal distribution is at a medium position.

## 5 DISCUSSION

In this paper, we introduce variational importance sampling (VIS), a novel method for efficiently learning parameters in latent variable models, based on the forward $\chi^2$ divergence. Unlike variational inference (VI), which maximizes the evidence lower bound (ELBO), VIS directly estimates and maximizes the marginal log-likelihood to learn model parameters. Our analyses demonstrate that the quality of the estimated marginal log-likelihood is assured with a large number of Monte Carlo samples and an optimal proposal distribution characterized by a small forward $\chi^2$ divergence. This highlights the statistical significance of choosing the proposal distribution. Experimental results across three different models validate VIS's ability to achieve both a higher marginal log-likelihood and a better parameter estimation. This underscores VIS as a promising learning method for addressing complex latent variable models. Nevertheless, it is worth noting that while this choice of the proposal distribution is statistically optimal for importance sampling, its practical significance in certain real-world applications might require further investigation and validation.

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

## A    APPENDIX

### A.1    GRADIENT ESTIMATORS IN THE VARIATIONAL INFERENCE

The derivative of $\text{ELBO}(\boldsymbol{x}; \theta, \phi)$ w.r.t. $\theta$ is estimated by

$$
\begin{aligned}
\frac{\partial \text{ELBO}(\boldsymbol{x}; \theta, \phi)}{\partial \theta} &= \int \frac{\partial \ln p(\boldsymbol{x}, \boldsymbol{z}; \theta)}{\partial \theta} q(\boldsymbol{z}|\boldsymbol{x}; \phi) \, \mathrm{d}\boldsymbol{z} \\
&\approx \frac{1}{K} \sum_{k=1}^{K} \frac{\partial \ln p\left(\boldsymbol{x}, \boldsymbol{z}^{(k)}; \theta\right)}{\partial \theta} \\
&= \frac{\partial}{\partial \theta} \frac{1}{K} \sum_{k=1}^{K} \ln p\left(\boldsymbol{x}, \boldsymbol{z}^{(k)}; \theta\right).
\end{aligned}
\tag{16}
$$

For the derivative of $\text{ELBO}(\boldsymbol{x}; \theta, \phi)$ w.r.t. $\phi$ at $\phi_0$, the score function gradient estimator is

$$
\begin{aligned}
\frac{\partial \text{ELBO}(\boldsymbol{x}; \theta, \phi)}{\partial \phi} &= \int \left[\ln p(\boldsymbol{x}, \boldsymbol{z}; \theta) - \ln q(\boldsymbol{z}|\boldsymbol{x}; \phi_0)\right] \frac{\partial q(\boldsymbol{z}|\boldsymbol{x}; \phi)}{\partial \phi} - q(\boldsymbol{z}|\boldsymbol{x}; \phi_0) \frac{\partial \ln q(\boldsymbol{z}|\boldsymbol{x}; \phi)}{\partial \phi} \, \mathrm{d}\boldsymbol{z} \\
&= \int \left[\ln p(\boldsymbol{x}, \boldsymbol{z}; \theta) - \ln q(\boldsymbol{z}|\boldsymbol{x}; \phi_0)\right] q(\boldsymbol{z}|\boldsymbol{x}; \phi_0) \frac{\partial \ln q(\boldsymbol{z}|\boldsymbol{x}; \phi)}{\partial \phi} \, \mathrm{d}\boldsymbol{z} \\
&\quad - \frac{\partial}{\partial \phi} \int q(\boldsymbol{z}|\boldsymbol{x}; \phi) \, \mathrm{d}\boldsymbol{z} \\
&\approx \frac{1}{K} \sum_{k=1}^{K} \left[\ln p\left(\boldsymbol{x}, \boldsymbol{z}^{(k)}; \theta\right) - \ln q\left(\boldsymbol{z}^{(k)} \big| \boldsymbol{x}; \phi_0\right)\right] \frac{\partial \ln q\left(\boldsymbol{z}^{(k)} \big| \boldsymbol{x}; \phi\right)}{\partial \phi} - 0 \\
&= \frac{\partial}{\partial \phi} \frac{-1}{2K} \sum_{k=1}^{K} \left[\ln p\left(\boldsymbol{x}, \boldsymbol{z}^{(k)}; \theta\right) - \ln q\left(\boldsymbol{z}^{(k)} \big| \boldsymbol{x}; \phi\right)\right]^2.
\end{aligned}
\tag{17}
$$

When the parameterization trick can be utilized, $\boldsymbol{z}|\boldsymbol{x}; \phi = g(\boldsymbol{\epsilon}|\boldsymbol{x}; \phi)$ where $\boldsymbol{\epsilon} \sim r(\boldsymbol{\epsilon})$, then

$$
q(\boldsymbol{z}|\boldsymbol{x}; \phi) \, \mathrm{d}\boldsymbol{z} = r(\boldsymbol{\epsilon}) \, \mathrm{d}\boldsymbol{\epsilon}.
\tag{18}
$$

Now, we can get the pathwise gradient estimator,

$$
\begin{aligned}
\frac{\partial \text{ELBO}(\boldsymbol{x}; \theta, \phi)}{\partial \phi} &= \frac{\partial}{\partial \phi} \int q(\boldsymbol{z}|\boldsymbol{x}; \phi) \left[\ln p(\boldsymbol{x}, \boldsymbol{z}; \theta) - \ln q(\boldsymbol{z}|\boldsymbol{x}; \phi)\right] \, \mathrm{d}\boldsymbol{z} \\
&= \frac{\partial}{\partial \phi} \int r(\boldsymbol{\epsilon}) \left[\ln p(\boldsymbol{x}, g(\boldsymbol{\epsilon}|\boldsymbol{x}; \phi)) - \ln q(g(\boldsymbol{z}|\boldsymbol{x}; \phi)|\boldsymbol{x}; \phi)\right] \, \mathrm{d}\boldsymbol{\epsilon} \\
&\approx \frac{\partial}{\partial \phi} \frac{1}{K} \sum_{k=1}^{K} \left[\ln p\left(\boldsymbol{x}, g\left(\boldsymbol{\epsilon}^{(k)} \big| \boldsymbol{x}; \phi\right); \theta\right) - \ln q\left(g\left(\boldsymbol{\epsilon}^{(k)} \big| \boldsymbol{x}; \phi\right) \big| \boldsymbol{x}; \phi\right)\right].
\end{aligned}
\tag{19}
$$

### A.2    GRADIENT ESTIMATOR OF THE IMPORTANCE SAMPLING

The derivative of $\ln p(\boldsymbol{x}; \theta)$ w.r.t. $\theta$ at $\theta_0$ is estimated by

$$
\begin{aligned}
\frac{\partial \ln p(\boldsymbol{x}; \theta)}{\partial \theta} &= \frac{1}{p(\boldsymbol{x}; \theta_0)} \int \frac{\partial p(\boldsymbol{x}, \boldsymbol{z}; \theta)}{\partial \theta} \, \mathrm{d}\boldsymbol{z} \\
&= \frac{1}{p(\boldsymbol{x}; \theta_0)} \int p(\boldsymbol{x}, \boldsymbol{z}; \theta_0) \frac{\partial \ln p(\boldsymbol{x}, \boldsymbol{z}; \theta)}{\partial \theta} \, \mathrm{d}\boldsymbol{z} \\
&\approx \frac{1}{\hat{p}(\boldsymbol{x}; \theta_0)} \frac{1}{K} \sum_{k=1}^{K} \frac{p\left(\boldsymbol{x}, \boldsymbol{z}^{(k)}; \theta_0\right)}{q\left(\boldsymbol{z}^{(k)} \big| \boldsymbol{x}; \phi_0\right)} \frac{\partial \ln p\left(\boldsymbol{x}, \boldsymbol{z}^{(k)}; \theta\right)}{\partial \theta} \\
&= \frac{1}{\hat{p}(\boldsymbol{x}; \theta_0)} \frac{\partial}{\partial \theta} \frac{1}{K} \sum_{k=1}^{K} \exp\left[\ln p\left(\boldsymbol{x}, \boldsymbol{z}^{(k)}; \theta\right) - \ln q\left(\boldsymbol{z}^{(k)} \big| \boldsymbol{x}; \phi\right)\right] \\
&= \frac{1}{\hat{p}(\boldsymbol{x}; \theta_0)} \frac{\partial \hat{p}(\boldsymbol{x}; \theta)}{\partial \theta} = \frac{\partial \ln \hat{p}(\boldsymbol{x}; \theta)}{\partial \theta}.
\end{aligned}
\tag{20}
$$

Due to the appearance of $\hat{p}(\boldsymbol{x}; \theta_0)$ in the denominator, $\frac{\partial \ln \hat{p}(\boldsymbol{x}; \theta, \phi)}{\partial \phi}$ is a magnitude up-biased estimator of $\frac{\partial \ln p(\boldsymbol{x}; \theta)}{\partial \phi}$. However, the direction of $\frac{\partial \ln \hat{p}(\boldsymbol{x}; \theta, \phi)}{\partial \phi}$ is unbiased:

$$
\begin{aligned}
\mathbb{E}_q \left[ \frac{\partial \hat{p}(\boldsymbol{x}; \theta, \phi)}{\partial \theta} \right] =& \mathbb{E}_q \left[ \frac{1}{K} \sum_{k=1}^{K} \frac{1}{q\left(\boldsymbol{z}^{(k)} \middle| \boldsymbol{x}; \phi\right)} \frac{\partial p\left(\boldsymbol{x}, \boldsymbol{z}^{(k)}; \theta\right)}{\partial \theta} \right] \\
=& \mathbb{E}_q \left[ \frac{1}{q(\boldsymbol{z}|\boldsymbol{x}; \phi)} \frac{\partial p(\boldsymbol{x}, \boldsymbol{z}; \theta)}{\partial \theta} \right] = \int \frac{\partial p(\boldsymbol{x}, \boldsymbol{z}; \theta)}{\partial \theta} \, \mathrm{d}\boldsymbol{z} \\
=& \frac{\partial}{\partial \theta} \int p(\boldsymbol{x}, \boldsymbol{z}; \theta) \, \mathrm{d}\boldsymbol{z} = \frac{\partial p(\boldsymbol{x}; \theta)}{\partial \theta}.
\end{aligned}
\tag{21}
$$

## A.3 GRADIENT ESTIMATOR FOR UPDATING THE PROPOSAL DISTRIBUTION IN VIS

In this section, we derive the score function gradient estimator and the pathwise gradient estimator for minimizing the forward $\chi^2$ divergence, which is equivalent to minimizing $\ln V(\boldsymbol{x}; \theta, \phi)$ in Eq. 11.

First, we show the derivation of Eq. 11.

$$
\begin{aligned}
\ln V(\boldsymbol{x}; \theta, \phi) \approx& \ln \frac{1}{K} \sum_{k=1}^{K} \frac{p\left(\boldsymbol{x}, \boldsymbol{z}^{(k)}; \theta\right)^2}{q\left(\boldsymbol{z}^{(k)} \middle| \boldsymbol{x}; \phi\right)^2} \\
=& \operatorname{logsumexp} \left[ 2 \ln p\left(\boldsymbol{x}, \boldsymbol{z}^{(k)}; \theta\right) - 2 \ln q\left(\boldsymbol{z}^{(k)} \middle| \boldsymbol{x}; \phi\right) \right] - \ln K \\
=:& \ln \hat{V}(\boldsymbol{x}; \theta, \phi).
\end{aligned}
\tag{22}
$$

The score function gradient estimator of $\ln V(\boldsymbol{x}; \theta, \phi)$ in Eq. 11 is

$$
\begin{aligned}
\frac{\partial \ln V(\boldsymbol{x}; \theta, \phi)}{\partial \phi} =& \frac{1}{V(\boldsymbol{x}; \theta, \phi_0)} \int p(\boldsymbol{x}, \boldsymbol{z}; \theta)^2 \frac{\partial}{\partial \phi} \frac{1}{q(\boldsymbol{z}|\boldsymbol{x}; \phi)} \, \mathrm{d}\boldsymbol{z} \\
=& \frac{1}{V(\boldsymbol{x}; \theta, \phi_0)} \int - \frac{p(\boldsymbol{x}, \boldsymbol{z}; \theta)^2}{q(\boldsymbol{z}|\boldsymbol{x}; \phi_0)} \frac{\partial}{\partial \phi} \ln q(\boldsymbol{z}|\boldsymbol{x}; \phi) \, \mathrm{d}\boldsymbol{z} \\
\approx& \frac{1}{\hat{V}(\boldsymbol{x}; \theta, \phi_0)} \frac{1}{K} \sum_{k=1}^{K} - \frac{p\left(\boldsymbol{x}, \boldsymbol{z}^{(k)}; \theta\right)^2}{q\left(\boldsymbol{z}^{(k)} \middle| \boldsymbol{x}; \phi_0\right)^2} \frac{\partial \ln q\left(\boldsymbol{z}^{(k)} \middle| \boldsymbol{x}; \phi\right)}{\partial \phi} \\
=& \frac{1}{\hat{V}(\boldsymbol{x}; \theta, \phi_0)} \frac{1}{K} \sum_{k=1}^{K} \frac{1}{2} \frac{\partial}{\partial \phi} \exp \left[ 2 \ln p\left(\boldsymbol{x}, \boldsymbol{z}^{(k)}; \theta\right) - 2 \ln q\left(\boldsymbol{z}^{(k)} \middle| \boldsymbol{x}; \phi\right) \right] \\
=& \frac{\partial}{\partial \phi} \frac{1}{2} \ln \hat{V}(\boldsymbol{x}; \theta, \phi).
\end{aligned}
\tag{23}
$$

When the reparameterization trick can be utilized, $\boldsymbol{z}|\boldsymbol{x}; \phi = g(\boldsymbol{\epsilon}|\boldsymbol{x}; \phi)$ where $\boldsymbol{\epsilon} \sim \boldsymbol{r}(\boldsymbol{\epsilon})$, then we have the transformation $q(\boldsymbol{z}|\boldsymbol{x}; \phi) \, \mathrm{d}\boldsymbol{z} = r(\boldsymbol{\epsilon}) \, \mathrm{d}\boldsymbol{\epsilon}$ (Schulman et al., 2015). Then,

$$
\begin{aligned}
\frac{\partial \ln V(\boldsymbol{x}; \theta, \phi)}{\partial \phi} =& \frac{1}{V(\boldsymbol{x}; \theta, \phi_0)} \frac{\partial}{\partial \phi} \int q(\boldsymbol{z}|\boldsymbol{x}; \phi) \frac{p(\boldsymbol{x}, \boldsymbol{z}; \theta)^2}{q(\boldsymbol{z}|\boldsymbol{x}; \phi)^2} \, \mathrm{d}\boldsymbol{z} \\
=& \frac{1}{V(\boldsymbol{x}; \theta, \phi_0)} \frac{\partial}{\partial \phi} \int r(\boldsymbol{\epsilon}) \frac{p(\boldsymbol{x}, \boldsymbol{z}; \theta)^2}{q(\boldsymbol{z}|\boldsymbol{x}; \phi)^2} \, \mathrm{d}\boldsymbol{\epsilon} \\
\approx& \frac{1}{V(\boldsymbol{x}; \theta, \phi)} \frac{\partial}{\partial \phi} \frac{1}{K} \sum_{k=1}^{K} \exp \left[ 2 \ln p\left(\boldsymbol{x}, \boldsymbol{z}^{(k)}; \theta\right) - 2 \ln q\left(\boldsymbol{z}^{(k)} \middle| \boldsymbol{x}; \phi\right) \right] \\
=& \frac{\partial}{\partial \phi} \ln \hat{V}(\boldsymbol{x}; \theta, \phi).
\end{aligned}
\tag{24}
$$

## A.4 LATENT MANIFOLD OF THE MNIST DATASET

The following figures are the latent manifolds of the MNIST dataset learned by different methods.

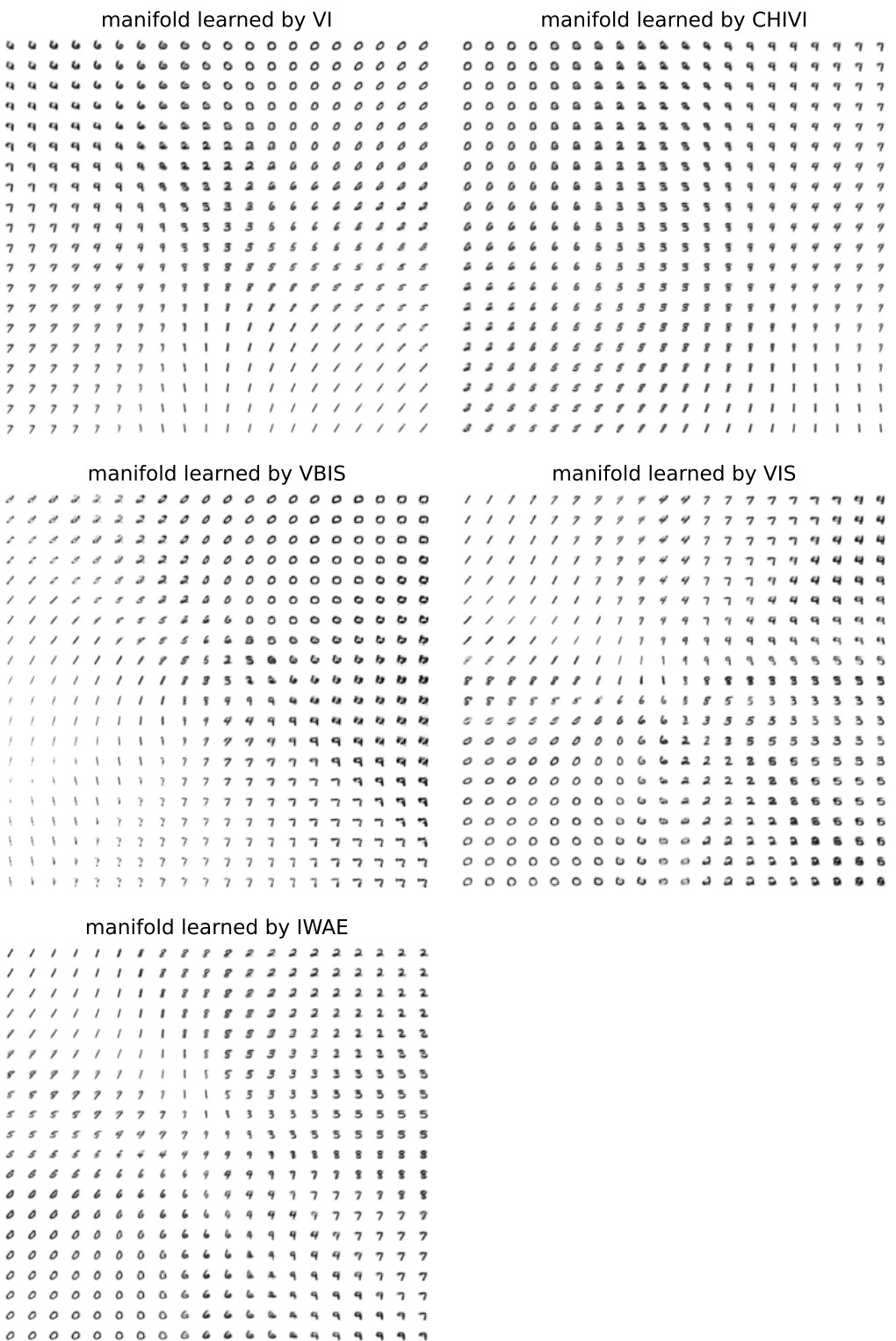

Figure 6: Latent manifolds of the MNIST dataset learned by different methods.

A.5 COMPARISON OF DIFFERENT GRADIENT ESTIMATORS OF EQ. 11

Considering that the numerical issue in minimizing the forward $\chi^2$ divergence is widely discovered by a lot of previous works (Pradier et al., 2019; Finke & Thiery, 2019; Geffner & Domke, 2020; Yao et al., 2018), we run VIS on the toy mixture model again (Sec. 4.1) using the {score function, pathwise} gradient estimator in {log, original} space for minimizing the forward $\chi^2$ divergence. Results in Fig. 7 show that the score function gradient estimator is better than the pathwise gradient estimator for minimizing the forward $\chi^2$ divergence. Besides, it is important to estimate it in log space so that the numerical stability of the score function gradient estimator can be promised.

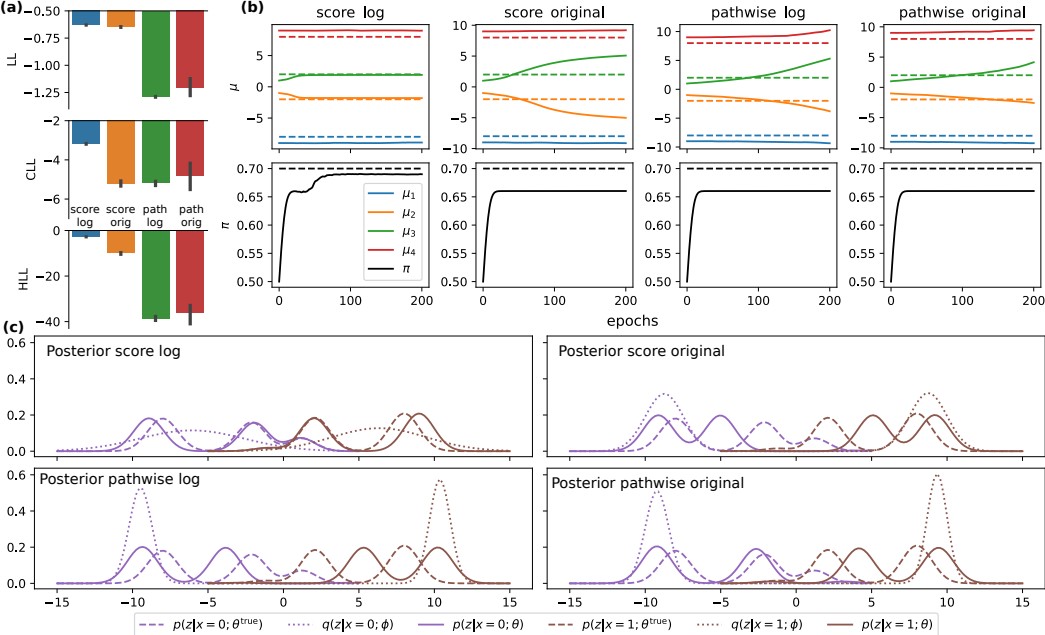

Figure 7: **(a)**: LL, CLL, and HLL evaluated on the test dataset. **(b)**: Convergence curves of the parameter set $\theta$ learned by different gradient estimators. The dashed curves are the true parameters used for generating the data, and the solid curves are the learned parameters. **(c)**: The posterior distribution given $x = 0$ and $x = 1$ learned by different gradient estimators. The dashed curves are the true posterior $p(\boldsymbol{z}|x;\theta^{\text{true}})$, the solid curves are the learned posterior $p(\boldsymbol{z}|x;\theta)$, and the dotted curves are the approximated posterior $q(z|x;\phi)$ learned in the variational/proposal distribution.

## A.6 RUNNING TIME OF DIFFERENT METHODS

Fig. 8 shows the test LL and corresponding running time of different methods w.r.t. the number of Monte Carlo $K$ on the synthetic POGLM dataset (Sec. 4.3). In general, the running times of all methods are linear to the number of Monte Carlo samples. With more Monte Carlo samples, all methods perform better, and VIS is consistently better than others especially when $K$ is large. When $K$ is small, all methods fail because of the complex nature of the POGLM problem. This implies that for complicated graphical models and high dimensional latent space, we do need enough Monte Carlo samples for all these sampling-based methods to become effective. Therefore, the number of Monte Carlo should be suitable to the complexity of the model/problem, rather than which method we choose.

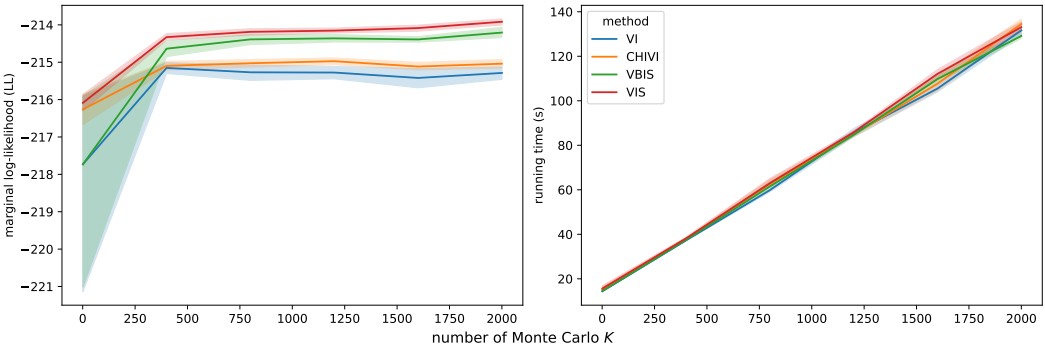

Figure 8: Test LL (left) and corresponding running time (right) of different methods w.r.t. the number of Monte Carlo $K$, on the synthetic POGLM dataset (Sec. 4.3).

## A.7 Forward KL divergence

(Jerfel et al., 2021) considers forward KL divergence as the target function for updating the proposal distribution since they noticed the drawback of the reverse KL divergence. According to (Sason & Verdú, 2016) and (Nishiyama & Sason, 2020), however, $\mathrm{KL}(p\|q)$ can be bounded by $\chi^2(p\|q)$:

$$\mathrm{KL}(p\|q) \leqslant \ln\left(1 + \chi^2(p\|q)\right) \leqslant \chi^2(p\|q), \tag{25}$$

but not vice versa. Therefore, minimizing the forward KL divergence might not be able to get the optimal proposal distribution, which should be obtained by minimizing the forward $\chi^2$ divergence. To validate this empirically, we compare minimizing the forward $\chi^2$ divergence (VIS) to minimizing the forward KL divergence (forward KL) on the toy mixture model again (Sec. 4.1), and the results are shown in Fig. 9.

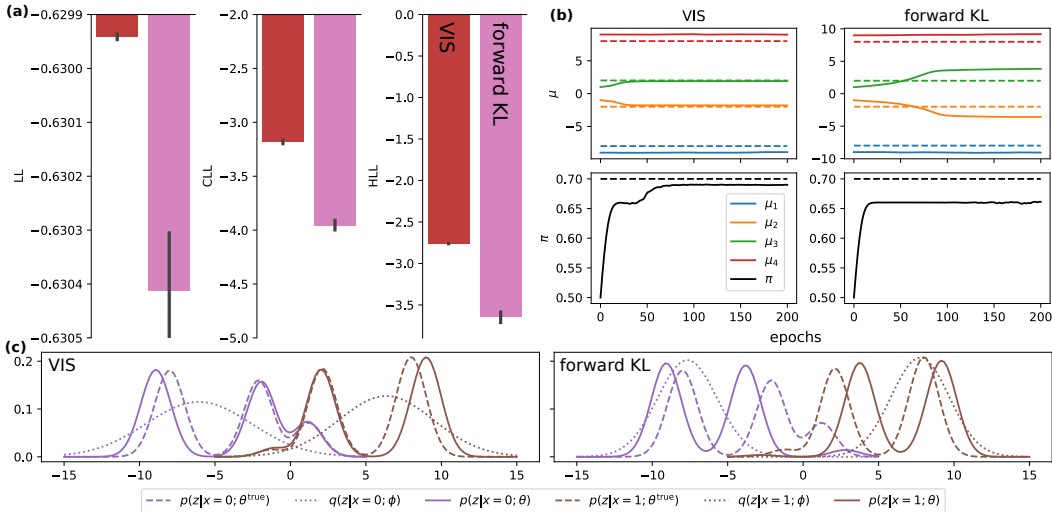

Figure 9: **(a)**: LL, CLL, and HLL evaluated on the test dataset. **(b)**: Convergence curves of the parameter set $\theta$ learned by VIS and forward KL. The dashed curves are the true parameters used for generating the data, and the solid curves are the learned parameters. **(c)**: The posterior distribution given $x = 0$ and $x = 1$ learned by different gradient estimators. The dashed curves are the true posterior $p(z|x; \theta^{\mathrm{true}})$, the solid curves are the learned posterior $p(z|x; \theta)$, and the dotted curves are the approximated posterior $q(z|x; \phi)$ learned in the variational/proposal distribution.

## A.8 Related works and contributions table

Here, we aim to offer a concise summary of our contributions and related works.

Table 1: Contributions.

| Contributions | Previous literatures |
| --- | --- |
| Motivate from the effectiveness of IS | [3] [5] [6] [7] |
| Aim at learning $\theta$ | [1] [3] [4] [5] [6] [7] |
| No restrictions on the $q$ distribution families | [1] [2] [3] |
| Directly minimizing forward $\chi^2$ divergence without surrogate | [2] [3] [5] [7] |
| Motivate from the bias of IS in log space | |
| Numerically stable gradient estimator in log space | |
| Extensive experiments on cases where no explicit decomposition $p(\boldsymbol{x}, \boldsymbol{z}; \theta) = p(\boldsymbol{x}\|\boldsymbol{z}; \theta)p(\boldsymbol{z}; \theta)$ exists | |
| Visualization for inferred latent and parameter $\theta$'s recovery | |

- **Motivate from the bias of IS in log space**: We start by comparing the bias of the $\ln \hat{p}(\boldsymbol{x}; \theta, \phi)$ and $\widehat{\mathrm{ELBO}}(\boldsymbol{x}; \theta, \phi)$ to analyze why doing IS and the optimal way of doing IS. And the conclusion about minimizing the forward $\chi^2$ divergence coincides with improving the effectiveness of the IS estimator (Fig. 1).
- **Numerically stable gradient estimator in log space**: Previous work already derived the gradient estimator for minimizing the $\chi^2$ divergence in the original space but not in log space. This leads to the numerical instability issue and scaling to the high dimensionality issue. We argue that it is critical to estimate its gradient in log space to obtain a numerically stable and succinct form of the gradient estimator, especially for the score function estimator (Fig. 7).
- **Extensive experiments on cases where no explicit decomposition** $p(\boldsymbol{x}, \boldsymbol{z}; \theta) = p(\boldsymbol{x}\|\boldsymbol{z}; \theta)p(\boldsymbol{z}; \theta)$ **exists**: Most of the previous work only do experiments on generative models with explicit decomposition $p(\boldsymbol{x}, \boldsymbol{z}; \theta) = p(\boldsymbol{x}\|\boldsymbol{z}; \theta)p(\boldsymbol{z}; \theta)$, unlike the POGLM. However, when such an explicit decomposition does not exist or when the generative posterior distribution $p(\boldsymbol{z}\|\boldsymbol{x}; \theta)$ and the approximating posterior distribution $q(\boldsymbol{z}\|\boldsymbol{x}; \phi)$ are not Gaussian, ELBO cannot be reformulated as $\mathrm{ELBO}(\boldsymbol{x}; \theta, \phi) = \mathbb{E}_q[\ln p(\boldsymbol{x}\|\boldsymbol{z}; \theta)] - \mathrm{KL}(q(\boldsymbol{z}\|\boldsymbol{x}; \phi)\|p(\boldsymbol{z}; \theta))$, and hence ELBO lost a lot of advantages. Therefore, we do need a variety of graphical models to understand the performance of different methods.
- **Visualization for inferred latent and parameter $\theta$'s recovery**: Although theoretical materials in this paper have shown the superiority of VIS, practical visualization of the behavior of different methods is still necessary for us to get an intuition of how and why VIS performs better than others.

[1] Burda et al. (2015)
[2] Dieng et al. (2017)
[3] Finke & Thiery (2019)
[4] Jerfel et al. (2021)
[5] Domke & Sheldon (2018)
[6] Su & Chen (2021)
[7] Akyildiz & Míguez (2021)

