# OpenReview forum: "Forward $\chi^2$ Divergence Based Variational Importance Sampling"
_ICLR.cc/2024/Conference — ICLR 2024 spotlight_

### Official Review · Reviewer_T2mb · 2023-10-29

**Soundness:** 4 excellent
**Presentation:** 4 excellent
**Contribution:** 3 good
**Rating:** 8
**Confidence:** 4

**Summary:**

This work proposes a new method of variational inference (VI) for latent variable models based on importance sampling, and shows its effectiveness with numerical experiments on toy model (synthetic), auto encoder model (real) and spike neural network (synthetic and real).

The method improves the tightness of the conventional VI by increasling the batch size of monte-carlo sampling, while giving up the exact computation/minimization of the approximation gap introduced with VI (cf. Eq. 8 and 11). This theoretical trade-off turned out to be beneficial in the experiments.

**Strengths:**

The paper is original and well written. The performance gain obtained with the proposed method is significant, which is one of the main contribution of the paper.

**Weaknesses:**

There is no discussion on the limitation of the proposed method.
From what I understood from the paper, I expect the following potential drawbacks:
  - Increase in training time based on the additional sampling
  - Instability in training due to the biased approximation (due to Eq. 8 and 11).

Detailed discussions on this matter are highly welcomed.

**Questions:**

The ground truth model of the toy experiment (Sec. 4.1) seems unidentifiable. In particular, the distribution of the hidden variable has multiple modes, which seems impossible to be identified only with the binary visible variable. This observation also explains well that despite the fact that the conventional VI substantially failed to estimate the modes, there is only a tiny difference in log-likelihood compared to other methods (order of 1e-4, see Fig. 2a). So, the question is, **why do you compare the parameter convergence, HLL and CLL**, which I think are meaningful to compare only if the model is identifiable? A similar argument also applies to the synthetic experiment of spike NN.

---

> ### Author Response · Authors · 2023-11-14
> **Reply to T2mb (1/2)**
>
> Dear Reviewer T2mb,
>
> Thank you very much for your time and valuable comments on our paper. We highly appreciate your recognition of the strengths of our paper. Hopefully, the following responses could resolve most of your concerns and answer your questions.
>
> ### Weaknesses
> **Discussion on limitation of VIS**: The last sentence of the discussion section does point out a limitation of VIS. "Nevertheless, it is worth noting that while this choice of proposal distribution in VIS is statistically optimal for importance sampling, its practical significance in certain real-world applications might require further investigation and validation." Therefore, the proposal distribution learned by our VIS method still functions as an approximated posterior distribution. It is closest to the true posterior in terms of forward $\chi^2$ divergence, which differs from the variational distribution learned by VI (maximizing ELBO) in terms of reverse KL divergence.
> * When the problem itself is complicated (e.g., complicated generative process and/or high latent dimensionality such as the POGLM model in Sec 4.3), a large number of Monte Carlo samples is indeed necessary, no matter for VIS or other baseline methods including VI. When working with VAE, people are already accustomed to using one Monte Carlo sample, but more Monte Carlo samples certainly improves the quality of $\widehat{\operatorname{ELBO}}$ since $\operatorname{ELBO}$ is also an integral expression w.r.t. $q(\boldsymbol z|\boldsymbol x;\phi)$. Anyway, however, employing a larger number of Monte Carlo samples always improves the estimation for all sampling-based estimation methods, including VI as depicted in Fig. 1. The main conclusion from Fig. 1 is that increasing the number of Monte Carlo increases the effectiveness of both $\widehat{\operatorname{ELBO}}$ and $\ln \hat p$, but it is only very helpful for tightening the bound of $\ln \hat p$ (with the speed of order $\frac 1 K$) rather than $\widehat{\operatorname{ELBO}}$. As we mentioned below Eq. 8, when the number of Monte Carlo samples $K=1$, $\widehat{\operatorname{ELBO}} = \ln \hat p$. Therefore, we need at least two Monte Carlo samples to distinguish them. **And using larger $K$ could lead to greater improvement to VIS than VI. This doesn't mean VIS requires larger $K$ to work or VIS takes more time (same $K$, then same sampling time for all methods including VIS and VI), so no extra training time.** In practice, for example, if two samples are enough for VI to converge, then two samples are also enough for VIS to converge, and VIS should still outperform others. I.e., the number of Monte Carlo should be suitable to the complexity of the model/problem, rather than which method you choose. In our experiments, we use the same number of Monte Carlo and enough Monte Carlo samples for a fair and insightful comparison of different methods, especially the learning behavior of different methods, for example in Sec 4.1 the toy mixture model (Fig. 2). **New experiment results regarding the number of Monte Carlo are shown in Appendix 6 and Fig. 8 in our latest revision.**
> * **Approximation of Eq. 8 and 11**:
>    * **Eq. 8**: Eq. 8 is deriving the bias of the numerical estimator of IS $\ln \hat p(\boldsymbol x;\theta,\phi)$, showing that its distance to the true $\ln \hat p(\boldsymbol x;\theta)$ is asymptotically much closer than $\widehat{\operatorname{ELBO}}(\boldsymbol x;\theta,\phi)$. This tells us one of the reasons why we should prefer to directly maximize $\ln \hat p(\boldsymbol x;\theta,\phi)$ rather than $\widehat{\operatorname{ELBO}}(\boldsymbol x;\theta,\phi)$. The $\approx$ in Eq. 8 is not related to the "biased approximation" since it is not an estimator.
>    * **Eq. 11**: It is a derivation of the numerically stable estimator of the forward $\chi^2$ divergence in log space. This quantifies the bias of the importance sampling (IS) estimator in log space and the effectiveness of the IS estimator. Hence, we minimize this to improve IS. It is likely that this approximation is biased, but the only way to approximate Eq. 10 $\ln V(\boldsymbol x|\theta,\phi)$ is to use such an approximation with samples $\boldsymbol z^{(k)}\sim q(\boldsymbol z|\boldsymbol x;\phi)$.

---

> > ### Author Response · Authors · 2023-11-14
> > **Reply to T2mb (2/2)**
> >
> > ### Questions
> > * **Identifiable**: The purpose of the toy example is to provide insights into the distinctions between VIS and other methods. Therefore, we deliberately choose a restricted approximating posterior distribution family $q$ parameterized by $\phi$. This deliberate choice allows us to observe and comprehend the diverse behaviors of different methods, facilitating a comparative analysis to determine their efficacy through visualization.
> > * **Identifiable of latent does not affect us to compare parameter recovery**: It is not critical to require the latent to be identifiable. Unidentifiable approximating posterior distribution $q$ can still serve as variational/proposal distribution for learning $\theta$. In most cases, we don't know the true posterior distribution family, and we are not trying to find a better approximating posterior distribution family for a particular model. The purpose of VIS is to find the optimal proposal distribution to do IS and to learn $\theta$ under any arbitrarily given approximating posterior distribution family. So, we can still compare the learning results of different methods.
> > * **Log-likelihood**: The magnitude of the log-likelihood is highly contingent on the model itself $p(\boldsymbol x;\theta) = \int p(\boldsymbol x,\boldsymbol z;\theta)\ \mathrm{d} \boldsymbol z$. The error bars presented in Fig. 2a statistically validate that VIS is significantly better than others for the task of learning the model parameter set $\theta$.
> > * **Parameter convergence**: Our aim is to utilize parameter convergence as a reliable confirmation that improved test log-likelihood aligns consistently with better parameter estimation.
> > * **HLL and CLL serve as two supplementary metrics**, aiding our understanding of why the inferred latent is better or worse, regardless of whether the latent is identifiable. Similar to LL, for example, **HLL informs us about the performances of various approximating posterior distributions $q(\boldsymbol z|\boldsymbol x;\phi)$, no matter the approximating posterior distribution family $q(\boldsymbol z|\boldsymbol x;\phi)$ is able to match the true posterior distribution family $p(\boldsymbol z|\boldsymbol x;\theta)$ or not**.
> > * In a real-world experiment, we will never know what is the distribution family of the generative model and the posterior distribution, but we still use marginal/hidden log-likelihood to compare the performance of different methods.
> >
> > We hope that this response addresses the majority of your concerns and questions, aiding in a clearer understanding of our paper. Your feedback is valuable, so please don't hesitate to provide additional comments or ask further questions.

---

### Official Review · Reviewer_QxSX · 2023-10-29

**Soundness:** 4 excellent
**Presentation:** 3 good
**Contribution:** 3 good
**Rating:** 8
**Confidence:** 3

**Summary:**

The paper focuses on the problem of approximate Bayesian inference in latent variable models and challenges the common wisdom of approximating the posterior with variational inference and maximizing the evidence lower bound (ELBO). The authors identify a mismatch between maximizing ELBO and maximizing loglikelihood, and posit that the latter is a better objective, as it is divorced from the quality of the approximate posterior. To tackle the maximization of loglikelihood, the paper proposes variational importance sampling (VIS), which maximizes the marginal directly via importance sampling. Upon inspecting the variance of the IS estimator, the paper also proposes minimizing the chi-squared divergence w.r.t. the approximate posterior.

The paper inspects the efficacy of this method on a series of latent variable models, and witness a consistent improvement on the inference of latent variable models.

**Strengths:**

I think the paper warrants acceptance due to the fact that it tackles a well-motivated problem with a simple, yet empirically effective method.

- Motivation: the maximization of ELBO as a proxy of likelihood maximization has long been a standard practice in latent variable, even though the mismatch can be significant. I agree with the authors that the mismatch should be inspected more carefully, and that one should separate the approximate posterior inference with the model itself.
- Methodology: the paper proposes a simple, yet elegant solution to the question of likelihood maximization, and motivate the reasoning behind the optimization of the chi-squared divergence from the perspective of the bias and variance of the IS estimator. This approach seems novel, even though I am not very up-to-date about the research in this regard.
- Experiments: the paper compares against alternative options in latent variable inference and showcase that the method can correctly infer the marginal likelihood, as well as outperforming competing methods in real-world datasets.

**Weaknesses:**

I am not entirely sure how the method proposed in this paper differs from previous work, as the practice of using Monte Carlo samples to sharpen variational bound is a topic that has been explored by previous work. I hope the authors can make some clarifications or have a small section in the manuscript that discusses the difference across different ways to combine IS with VI.

- It is neat to see the same chi-squared divergence on both the bias and effectiveness of the IS estimator, but eq. 8 contains an approximation that seems to work mostly for large $K$s, and the effectiveness is evaluated at the estimator $\hat{p}$ and not $\log \hat{p}$, making the connection seem a bit artificial. Is it possible to minimize a divergence in the form of, e.g., the first line in eq. 8?

**Questions:**

- The experiments presented in the paper show that VIS performs well, but it seems that a large number of Monte Carlo samples are chosen in many of the experiments. This seems that it can present a significant computational cost. Could the authors include some explanation about the effect of the number of Monte Carlo samples on the efficacy of VIS, and clarify if taking many Monte Carlo samples is practical in training latent variable models?

---

> ### Author Response · Authors · 2023-11-14
> **Reply to QxSX (1/2)**
>
> Dear Reviewer QxSX,
>
> Thank you very much for your time and valuable comments on our paper. We highly appreciate your recognition of the strengths of our paper. Hopefully, the following responses could resolve most of your concerns and answer your questions.
>
> ### Weaknesses
> In the third paragraph of Section 1, various methods of combining IS with VI are briefly mentioned. Additionally, these methods are summarized as baseline techniques at the beginning of Sec 4. Below, we aim to provide a more detailed explanation. In the context of a latent variable model $p(\boldsymbol x, \boldsymbol z;\theta)$, two main challenges arise: the learning problem, which focuses on learning $\theta$, and the inference problem, which aims to find an approximating posterior distribution $q(\boldsymbol z|\boldsymbol x;\phi)$ that approximates $p(\boldsymbol z|\boldsymbol x;\theta)$ for a specific $\theta$.
> * Learning $\theta$: The process of learning $\theta$ involves maximizing the log-likelihood, denoted as $\arg\max_\theta \ln p(\boldsymbol x;\theta)$ in log space. Two distinct optimization approaches exist for this task:
>    * (a) Maximizing $\widehat{\operatorname{ELBO}}(\boldsymbol x;\theta,\phi)$ w.r.t. to $\theta$, which is a hard lower-bound of $\ln p(\boldsymbol x;\theta)$, measured by $\operatorname{KL}(q\\|p)$.
>    * (b) Maximizing $\ln \hat p(\boldsymbol x;\theta,\phi)$ w.r.t. $\theta$, which is an asymptotically tighter lower-bound of $\ln p(\boldsymbol x;\theta)$, measured by $\chi^2(p\\|q)/(2K)$.
>
> Although both of them are sampling-based methods, the former one is usually called VI, and the latter one is usually called IS. Both of them requires sampling $K$ samples $\left\\{\boldsymbol z^{(k)}\right\\}_{k=1}^K$ from the current approximating posterior distribution $q(\boldsymbol z|\boldsymbol x;\phi)$.
> * Learning $\phi$: The ways to learn $\phi$ under a fixed $\theta$ are various. For example:
>    * (1) Minimizing the reverse KL divergence $\operatorname{KL}(q(\boldsymbol z|\boldsymbol x;\phi)\\|p(\boldsymbol z|\boldsymbol x;\phi))$ which happen to also maximizing $\widehat{\operatorname{ELBO}}(\boldsymbol x;\theta,\phi)$ but w.r.t. $\phi$.
>    * (2) Minimizing the forward $\chi^2$ divergence $\chi^2(p(\boldsymbol z|\boldsymbol x;\theta)\\|q(\boldsymbol z|\boldsymbol x;\phi))$
>    * (3) $\alpha$ divergence-based methods or forward KL divergence
>
> Subsequently, we can comprehend the various ways of combining them:
> * VI: (a)*(1). Since they utilize the same target function $\widehat{\operatorname{ELBO}}$.
> * CHIVI: (a)*((1) + (2)). Since they assume squeezing the posterior by a lower-bound ELBO and an upper-bound CUBO could improve the posterior approximating, but this may not be the best approximating posterior for learning $\theta$.
> * VBIS: (b)*(1). They assume the variational distribution found by VI (1) could serve as a proposal distribution for learning $\theta$ (a).
>
> **For your questions regarding $K$**, please see the following **Questions** part.
>
> **Regarding your inquiry about Eq. 8**, it's challenging to analytically derive the effectiveness (variance) of the estimator $\ln \hat p(\boldsymbol x;\theta,\phi)$ in log space. However, empirical evidence from Fig. 1 demonstrates that reducing the variance in the original space also leads to a reduction in variance in the log space.
>
> **Minimizing the first line in Eq. 8**: We argue that this is not practical, because the first line of Eq. 8 includes samples. Eq. 8 is deriving an estimator's bias. Bias is one of the metrics to evaluate a statistic, so its expression cannot include concrete samples. Mathematically we have to eliminate concrete samples $\boldsymbol z^{(k)}$ from the expression to obtain a sample-free analytical form of the bias, i.e. the last line of Eq. 8.

---

> > ### Author Response · Authors · 2023-11-14
> > **Reply to QxSX (2/2)**
> >
> > ### Questions
> > When the problem itself is complicated (e.g., complicated generative process and/or high latent dimensionality such as the POGLM model in Sec 4.3), a large number of Monte Carlo samples is indeed necessary, no matter for VIS or other baseline methods including VI. When working with VAE, people are already accustomed to using one Monte Carlo sample, but more Monte Carlo samples certainly improves the quality of $\widehat{\operatorname{ELBO}}$ since $\operatorname{ELBO}$ is also an integral expression w.r.t. $q(\boldsymbol z|\boldsymbol x;\phi)$. Anyway, however, employing a larger number of Monte Carlo samples always improves the estimation for all sampling-based estimation methods, including VI as depicted in Fig. 1. The main conclusion from Fig. 1 is that increasing the number of Monte Carlo increases the effectiveness of both $\widehat{\operatorname{ELBO}}$ and $\ln \hat p$, but it is only very helpful for tightening the bound of $\ln \hat p$ (with speed of order $\frac 1 K$) rather than $\widehat{\operatorname{ELBO}}$. As we mentioned below Eq. 8, when the number of Monte Carlo samples $K=1$, $\widehat{\operatorname{ELBO}} = \ln \hat p$. Therefore, we need at least two Monte Carlo samples to distinguish them. **And using larger $K$ could lead to greater improvement to VIS than VI. This doesn't mean VIS requires larger $K$ to work or VIS takes more time (same $K$, then same sampling time for all methods including VIS and VI), so no extra training time.** In practice, for example, if two samples are enough for VI to converge, then two samples are also enough for VIS to converge, and VIS should still outperform others. I.e., the number of Monte Carlo should be suitable to the complexity of the model/problem, rather than which method you choose. In our experiments, we use the same number of Monte Carlo and enough Monte Carlo samples for a fair and insightful comparison of different methods, especially the learning behavior of different methods, for example in Sec 4.1 the toy mixture model (Fig. 2). **New experiment results regarding the number of Monte Carlo are in Appendix 6 and Fig. 8 in our latest revision.**
> >
> > We hope that this response addresses the majority of your concerns and questions, aiding in a clearer understanding of our paper. Your feedback is valuable, so please don't hesitate to provide additional comments or ask further questions.

---

### Official Review · Reviewer_J8Yz · 2023-10-31

**Soundness:** 3 good
**Presentation:** 2 fair
**Contribution:** 2 fair
**Rating:** 6
**Confidence:** 4

**Summary:**

This work proposes an adaptive importance-sampling algorithm which seeks to improve the proposal distribution by minimising the chi-square divergence from the target distribution to the proposal.

**Strengths:**

There are a large number of experiments. The proposed method seems to perform well.

Approximating the gradient of the logarithm of the forward $\chi^2$-divergence (rather than the gradient of the $\chi^2$-divergence) seems to enhance numerical stability and seems novel. Though it should be noted that existing approaches improve numerical stability by minimising not the forward $\chi^2$-divergence itself but rather the forward $\chi^2$-divergence multiplied by the squared normalising constant of the target distribution: $p_\theta(x)^2$ (see, e.g., [2]). It is not clear how this compares to the log-space approach taken here.

[2] Akyildiz, Ö. D., & Míguez, J. (2021). Convergence rates for optimised adaptive importance samplers. Statistics and Computing, 31, 1-17.

**Weaknesses:**

Targetting the $\chi^2$-divergence as an objective for improving the proposal distribution in adaptive importance sampling is not novel. The fact that minimising the variance of the importance weights is equivalent to minimising the chi-square divergence from the target to the proposal is well known in the importance-sampling literature and has already often been used to improve proposal distributions within adaptive importance-sampling schemes, e.g. [1, 2] and references therein.

Furthermore, using such adaptive-importance-sampling approaches for variational inference is already extensively discussed in [3].:
1. Algorithm 1 of the present paper is a special case of the generic method described in [3, Section 3] (in particular, see [3, Section 3.4]);
2. the $\theta$-gradient from Equation 6 of the present work is already well known (see, e.g. [3]).
3. However, from the author's rebuttal, it is now more clear to me that their $\phi$ gradient is slightly different than in [3] because they derive the $\phi$-gradient in log-space.

[1] Jona‐Lasinio, G., Piccioni, M., & Ramponi, A. (1999). Selection of importance weights for monte carlo estimation of normalizing constants. Communications in Statistics-Simulation and Computation, 28(2), 441-462.

[2] Akyildiz, Ö. D., & Míguez, J. (2021). Convergence rates for optimised adaptive importance samplers. Statistics and Computing, 31, 1-17.

[3] Finke, A., & Thiery, A. H. (2019). On importance-weighted autoencoders. arXiv preprint arXiv:1907.10477.

**Questions:**

How does the approach compare with replacing the forward $\chi^2$- with forward KL-divergence (i.e., effectively two out of the three "phases" of reweighted wake--sleep)?

**Details Of Ethics Concerns:**

no concerns

---

> ### Author Response · Authors · 2023-11-14
> **Reply to J8Yz**
>
> Dear Reviewer J8Yz,
>
> Thank you very much for your time and valuable comments on our paper. We would like to make clarifications on your valuable suggestions.
>
> * [1] Jona-Lasinio et al. (1999): We acknowledge the presence of existing works that employ adaptive strategies to minimize the forward $\chi^2$ divergence, aiming to reduce the variance of the IS estimator. However, this paper only provides a general guideline but not a concrete algorithm for achieving the goal of minimizing the forward $\chi^2$ divergence. Additionally, our paper introduces another perspective on minimizing the forward $\chi^2$ divergence by considering the bias of $\ln \hat p(\boldsymbol x;\theta,\phi)$ in log space.
> * [2] Akyildiz et al. (2021): This paper primarily discusses minimizing $\chi^2$ divergence for distributions from the exponential distribution family. In contrast, our gradient estimator is not restricted to such a distribution family; it is applicable to all distributions, as demonstrated in the toy model (Sec 4.1) and the POGLM model (Sec 4.3). The true posterior distribution family and the approximated distribution family in these two models are way more complicated than the exponential distribution family, but we can still make VIS work for these arbitrary distribution families.
> * [3] Finke et al. (2019): While this paper addresses the gradient breakdown problem in the IWAE (Burda et al., 2016) paper and by providing a correct gradient estimator for optimizing the encoder or the proposal distribution, it's still important to highlight differences. Firstly, their gradient estimator is in the original space (Dieng et al., 2017), unlike ours, which is in log space. This is nontrivial and an important way to make the gradient estimator numerically stable. **Please check Appendix 5 and Fig 7 in our latest revision for insights into the numerical stability issue.** Besides, their experiments are confined to cases where the generative model has an explicit decomposition of $p(\boldsymbol x, \boldsymbol z;\theta) = p(\boldsymbol x|\boldsymbol z;\theta) p(\boldsymbol z;\theta)$. In contrast, our POGLM model in Sec 4.3 extends the analysis to a complex and general graphical model where such an explicit decomposition does not exist. Even for such a complicated general graphical model, our VIS can still work and outperform other baselines, which proves the practical effectiveness of our method.
>
> Furthermore, to the best of our knowledge, there is a gap in the existing literature concerning extensive experiments that systematically compare the practical learning performances of various methods across a diverse range of latent variable models. We believe it is necessary to visualize the behaviors of different methods (e.g. the toy model in Section 4.1 and the POGLM model in Sec 4.3) for readers to fully understand and convince how VIS works better than others.
>
> In conclusion, we respectfully hold a different perspective, asserting that our work indeed brings novelty and valuable contributions. We hope that our response has addressed your concerns and provided clarity regarding the significance of our research.

---

> ### Comment · Reviewer_J8Yz · 2023-11-18
>
> Thank you for the clarification.
>
> 1. I agree that it makes sense to minimise the chi-square divergence (or the following quantity that is proportional to it for fixed $\theta$)
> $$E_{z \sim q_\phi(z|x)}\Bigl[\Bigl(\frac{p_\theta(x, z)}{q_\phi(z|x)}\Bigr)^2\Bigr]$$
> in log-space. And you're right, this makes your $\phi$-gradient slightly different from some existing adaptive-importance sampling settings (the $\theta$-gradient is standard). I will raise my score.
> 2. Yes, [2] assumes exponential families. I don't think this specific reference needs to necessarily be cited in the paper. Rather, along with [1], I only gave it as an example of the fact that minimising the variance of the importance weights (equivalently, minimising the above-mentioned chi-square divergence) is a well-known technique in the importance-sampling literature.
> 3. This context needs to be made more clear in the paper, e.g., in the introduction.
>
> There are still some typos in the revised manuscript. I haven't re-read everything carefully but noticed that Eq. 12 should probably be "$\approx$" rather than "$=$"

---

> > ### Comment · Reviewer_J8Yz · 2023-11-18
> >
> > I have raised my score to 5. I would be willing to raise if further if the authors can demonstrate that their approach works better than simply using the forward KL-divergence (i.e., this would be nothing other than Reweighted Wake--Sleep without the "sleep" phase which is often not used).

---

> ### Author Response · Authors · 2023-11-18
> **New experiment results regarding forward KL**
>
> Dear Reviewer J8Yz
>
> We would like to express our sincere thanks for your acknowledgment of our paper and your new score.
> * **Regarding [2]**.
>     * Eq. 10 in our paper shows that $\chi^2(p\\|q) = \frac{1}{p(\boldsymbol x;\theta)^2} V(\boldsymbol x;\theta,\phi) - 1$. So, our target function $V(\boldsymbol x;\theta;\phi) = p(\boldsymbol x;\theta)^2\chi^2(p\\|q) + p(\boldsymbol x;\theta)^2$ in original space is the same as existing works.
>     * Theoretically, $p(\boldsymbol x;\theta)^2$ is a constant w.r.t. $\phi$. So, we (and existing works) drop the term of $p(\boldsymbol x;\theta)^2$, to only minimize $V(\boldsymbol x;\theta,\phi) = \int \frac{p(\boldsymbol x,\boldsymbol z;\theta)^2}{q(\boldsymbol z|\boldsymbol x;\phi)}\ \mathrm d \boldsymbol z$. We acknowledge that this step is helpful for numerical stability because the target function $V(\boldsymbol x;\theta,\phi)$ will not be affected by the magnitude of $p(\boldsymbol x;\theta)^2$.
>     * We take an additional step to perform the minimization in log space, which further improves the numerical stability significantly.
> * **Eq. 12**. Thanks, it is $\approx$. We will certainly do further proofreading to make sure there is no typo.
> * **Regarding [3]**. Thanks for this suggestion. We have added some new discussions to the introduction part. Due to the 9-page limit, some detailed texts are located in Appendix 8 (with a hyperlink directing from the introduction). Please check the latest revision.
> * **Comparing with forward KL**. Thanks for this suggestion. We do have the result in our hands. For example, Jerfel et al. (2021) also found the problem of the reverse KL divergence and investigated the forward KL divergence, but their algorithm is only confined to Gaussian distribution. **According to your suggestion, we had a new section in Appendix A.7 showing that forward $\chi^2$ divergence is better than forward KL divergence.** The reason should be from Sason et al. (2016) and Nishiyama et al. (2020) that: $\operatorname{KL}(p\\|q) \leqslant \chi^2(p\\|q)$. I.e., $\chi^2(p\\|q)$ can bound $\operatorname{KL}(p\\|q)$ but not vice versa. Therefore, minimizing forward KL divergence cannot promise a direct minimization of forward $\chi^2$ divergence.
>
> Please check Fig. 9 in Appendix 7 to see the new results. Thank you very much again for your valuable feedback on our initial response.
>
> New references:
> Sason, I., & Verdú, S. (2016). $ f $-divergence Inequalities. IEEE Transactions on Information Theory, 62(11), 5973-6006.
> Nishiyama, T., & Sason, I. (2020). On relations between the relative entropy and $χ^2$-divergence, generalizations and applications. Entropy, 22(5), 563.

---

> > ### Author Response · Authors · 2023-11-22
> > **Sincerely looking forward to your feedback before deadline**
> >
> > Dear Reviewer J8Yz,
> >
> > Thank you very much again for your initial review and the new score. Given it is very close to the discussion deadline, we would like to kindly remind you that we have included the result of the comparison with the forward KL requested by you last time. We are more than willing to hear your new feedback on it. We appreciate your time and invaluable feedback.
> >
> > Best,

---

### Official Review · Reviewer_84Pi · 2023-11-01

**Soundness:** 2 fair
**Presentation:** 2 fair
**Contribution:** 2 fair
**Rating:** 5
**Confidence:** 3

**Summary:**

The paper proposes a variational algorithm for learning model and latent parameters in a latent variable model which at each step first updates the variational distribution by minimizing forward chi-square divergence and then uses this distribution to estimate the log marginal likelihood using Importance Sampling. The optimization is done by gradient ascent where the gradients are estimated by MC sampling.
The proposed algorithm is compared against many contemporary algorithms on simulated and real world datasets, including a large scale case study on multi-neuron interaction modelled by a custom made partially observable GLM.

**Strengths:**

1. The paper tries well to motivate itself well and the use of chi-square divergence objective as means of finding the optimal distribution for IS is theoretically sound.
2. The paper has done experiments and analysis on may simulated and real world datasets. The experiment on GLP model for multi-neurns activation is well documented and insightful. Some of the plots look good and match the narrative.
3. The method proposed seems sound to me and the results show that it can perform better than contemporary VI algorithms on the tasks given in the paper.

**Weaknesses:**

1. The paper is not polished yet, although the major parts are all there, it may require another thorough pass.
it has too many mistakes and typos:, the notation changes from bold to normal in many places, the title has a typo: 'importane', 'log function is a convex function'.
2. Some of the references and recent literature is missing which have looked on the quality of different divergence objectives such as CUBO and ELBO for finding the optimal sampling distribution.
3. The theory part and the algorithm part can be emphasized more, right now it feels to compressed and dense. The figure 2 is good but it has too many colors and things to unpack, maybe use solid line for true posterior as I was thoroughly confused by the legend choices, and the use of two colors for showing modes reduced readibility for me atleast.
4. It is the bane of chi-square divergence methods that it  does not scale well with dimensions covered in the papers here: https://arxiv.org/pdf/2010.09541.pdf and https://arxiv.org/abs/1802.02538 and it seems that this method may not scale well as it uses Chi-squared divergence minimization.

**Questions:**

1. What is the dimensionality of the POGLM model, do the authors intend to use this method as a tool for low dimensional complex posteriors because both IS and CUBO do not scale well with dimensions and even large sample size as done in this paper will not help.
2. Maybe include this in your conclusion section and discuss this as a limitation ?
3. Did the authors use any other optimizers other than ADAM, did it have any effect, how did you choose the optimization algorithm hyperprameters like learning rate etc. ?
4. Did reparameterization gradients perform better than score gradients in the case where they both were available.

---

> ### Author Response · Authors · 2023-11-14
> **Reply to 84Pi (1/3)**
>
> Dear Reviewer 84Pi,
>
> Thank you very much for your time and valuable comments on our paper. We highly appreciate your recognition of the strengths of our paper. Hopefully, the following responses could resolve most of your concerns and answer your questions.
>
> ### Weaknesses:
> 1. Thank you for bringing attention to the typo in the title. We apologize for any confusion caused by these errors. Please check the polished latest revision.
>    **Notation**: In the method section, we employ vector forms $\boldsymbol x, \boldsymbol z$ for generality. However, in the experiments, we adapt them to the corresponding typefaces based on the model. For instance, in Sec 4.1 Toy example, both $x$ and $z$ are scalars, so we use the normal typeface. In Sec 4.3 POGLM, every observable $\boldsymbol X \in \mathbb N^{\text{time}\times \text{visible neurons}}$ and every latent $\boldsymbol Z \in \mathbb N^{\text{time}\times \text{hidden neurons}}$ are multivariate time sequences represented in matrix forms, so we use the capital bold typeface.
>    **log**: Thanks! log is a concave function.
> 2. **The papers related to CUBO and ELBO had been included in our initial submission. The second baseline method CHIVI (Dieng et al., 2017), explained at the beginning of Sec 4, is the CUBO-ELBO paper.** Specifically, they utilize CUBO as an upper-bound and ELBO as a lower-bound to squeeze the approximated posterior. However, all discussions in the CUBO paper revolve around the inference problem, i.e., approximating the posterior distribution $q(\boldsymbol z|\boldsymbol x;\phi)$, rather than determining the optimal proposal distribution for learning the parameter $\theta$ in $p(\boldsymbol x;\theta)$ as we were doing.
> 3. We apologize for the compressed and dense nature of the method section (Sec 3). Due to the 9-page limit, we acknowledge that crucial materials, such as related works, mathematical derivations (e.g., the gradient estimator), and detailed experiment explanations, may not have been extensively expanded in the main content. Most of the important lengthy derivations are provided in the appendix, allowing readers to verify the correctness of the result equations presented in the main content. Regarding Fig. 2, we appreciate the opportunity to resolve any confusion and are prepared to carefully explain all the details to you now.
>    **Fig 2**: To comprehensively understand why VIS outperforms other methods, it's crucial to visualize three distributions:
>    * the true posterior $p(z|x;\theta^{\mathrm{true}})$ (dashed)
>    * the learned posterior $p(z|x;\theta)$ (solid)
>    * the approximated (variational/proposal) posterior $q(z|x;\phi)$ (dotted).
>
>    **This linestyle assignment aligns with the convergence curves shown in Fig. 2B.** Given $x\in\\{0,1\\}$, we use two colors: purple for $z|x=0$ and brown for $z|x=1$. We can ignore brown curves first and only focus on purple curves to understand why VIS is better than others.
>    * **Conclusion 1**: the dotted curve of VIS is the widest, providing the widest effective sampling range, which is beneficial for learning $\theta$ through IS.
>    * **Conclusion 2**: the dotted curve (approximated posterior) and the solid curve (learned posterior) of VI exhibit the closest match, resulting in the smallest reverse KL divergence. However, the solid curve (learned posterior) is significantly distant from the dashed curve (true posterior). This suggests that maximizing ELBO in VI for learning $\theta$ could lead to a situation where the learned and approximated posteriors are close in reverse KL divergence (yielding a high ELBO), but both are far from the true posterior (resulting in a low marginal likelihood). This significantly impedes the learning of $\theta$.
> 4. **Scale to work for high dimensional latent space: This stands as one of our significant contributions.** The two papers you mentioned, along with Pradier et al. (2019) cited in our paper, acknowledge the instability associated with minimizing $\chi^2$ divergence, particularly in high-dimensional scenarios. **The lack of success in minimizing the $\chi^2$ divergence could be attributed to the numerically unstable form of their gradient estimator. In contrast, our gradient estimator (Equations 12 and 24) is derived in log space and presented in a simple and numerically stable form. Please check Appendix 5 and Fig 7 in our latest revision.** Our succinct gradient estimator makes VIS always work, even for the most general case that $p(\boldsymbol x,\boldsymbol z;\theta)$ cannot be explicitly decomposed into $p(\boldsymbol x|\boldsymbol z;\theta) p(\boldsymbol z;\theta)$. The POGLM experiments indeed confirm that VIS is robust across different models, numerically stable, and scalable to high latent dimensionality (addressing Question 1). Having successfully overcome the challenges in previous research related to $\chi^2$ divergence, we consider this a strength and a noteworthy contribution to our work, rather than a weakness.

---

> > ### Author Response · Authors · 2023-11-14
> > **Reply to 84Pi (2/3)**
> >
> > ### Questions
> > 1. This is for question 1 and 2
> >    * **The dimensionality of the latent space in POGLM** is the number of hidden neurons $\times$ the number of time bins, so it is a **high dimensional** problem.
> >    * **Scaling issue: Anyway, employing a larger number of Monte Carlo samples always improves the estimation for all sampling-based estimation methods, including VI as depicted in Fig. 1.** But when the problem itself is complicated, a large number of Monte Carlo samples is indeed necessary, no matter for VIS or other baseline methods including VI (ELBO). **Please check Appendix 6 and Fig 8 in our latest revision.**
> >    * Our main goal is learning $\theta$ to achieve high marginal log-likelihood. We are not studying dimensionality reduction. And we are not trying to find the best variational/proposal distribution family for a particular model. The POGLM experiment is to show that for such a complicated generative model, VIS obtains the most optimal approximated posterior, which serves for recovering the model parameter $\theta$ the best.
> > 2. /
> > 3.
> >    * **Optimizer**: We opt for ADAM as our choice of optimizer primarily because it has become the default preference across the board in current practices. While the selection of the optimizer falls outside the specific focus of this paper, we did experiment with Adagrad, and the comparative results did not exhibit significant changes. Notably, people commonly employ ADAM, Adagrad, or other optimizers to train VAE using VI, and the same applies to training VAE using our VIS. In the interest of a fair and effective comparison, we adhere to the widely used ADAM optimizer.
> >    * **Learning rate**: To determine the learning rate, we employ cross-validation, aligning with standard practices in many machine learning tasks. It's crucial to emphasize that the learning rate is a hyperparameter associated with the optimizer rather than being intrinsic to the method or the model. For the sake of a fair comparison, we set the learning rate to be consistent and, more importantly, suitable for the specific model and problem under consideration.
> > 4. Thanks for this good question. We also investigated the **differences between the score function estimator and the pathwise gradient estimator**, according to the discussion from Schulman et al. (2015). **A one-sentence conclusion is the score function estimator is more stable than the pathwise gradient estimator because of the expression form of the forward $\chi^2$ divergence.** This stability difference is highlighted as another crucial factor contributing to the historical instability observed in prior research on $\chi^2$. The score function gradient estimator of $\frac{\partial\operatorname{ELBO}}{\partial \phi}$ at $\phi_0$ is $\frac 1 K \sum_{k=1}^K \left[\ln p\left(\boldsymbol x,\boldsymbol z^{(k)};\theta\right) - \ln q\left(\boldsymbol z^{(k)} \middle| \boldsymbol x;\phi_0\right)\right] \frac{\partial \ln q\left(\boldsymbol z^{(k)}\middle|\boldsymbol x;\phi\right)}{\partial\phi}$ (Eq. 17), where $\phi_0$ appears outside of the derivative explicitly. The pathwise gradient estimator of $\frac{\partial\operatorname{ELBO}}{\partial \phi}$ at $\phi_0$ is $\frac 1 K \sum_{k=1}^K \left[\ln p\left(\boldsymbol x,g\left(\boldsymbol \epsilon^{(k)}\middle|\boldsymbol x;\phi\right);\theta\right) - \ln q\left(g\left(\boldsymbol \epsilon^{(k)}\middle|\boldsymbol x;\phi\right)\middle|\boldsymbol x;\phi\right)\right]$ (Eq. 19), where $\phi_0$ does not appear outside of the derivative explicitly. This makes the pathwise gradient estimator more stable than the score function gradient estimator for ELBO. However, it is another story for our forward $\chi^2$ divergence target function $\ln \hat V = \operatorname{logsumexp}\left[2\ln p\left(\boldsymbol x,\boldsymbol z^{(k)};\theta\right) - 2\ln q\left(\boldsymbol z^{(k)} \middle|\boldsymbol x;\phi\right)\right]$ (Eq. 11, Eq. 22). The score function gradient estimator of $\ln V$ w.r.t. $\phi$ is $\frac{\partial}{\partial \phi} \frac 1 2 \ln \hat V(\boldsymbol x;\theta,\phi)$, where $\phi_0$ already does not appear outside of the derivative explicitly. The pathwise gradient estimator of $\ln V$ w.r.t. $\phi$ is $\frac{\partial}{\partial \phi} \ln \hat V(\boldsymbol x;\theta,\phi)$, where $\phi_0$ also does not appear outside of the derivative explicitly. Therefore, for minimizing $\ln V$ (minimizing the forward $\chi^2$ divergence in the log space), there is no benefit from the expression of the gradient estimator, but changing from score function to pathwise additionally increases the path complexity to the parameter $\phi$. Therefore, up to our investigation, VIS falls into the case where the score function gradient estimator is better than the pathwise gradient estimator, at least for all experiments conducted in our paper. **Please check Appendix 5 and Fig 7 in our latest revision.**

---

> > > ### Author Response · Authors · 2023-11-14
> > > **Reply to 84Pi (3/3)**
> > >
> > > We hope that this response addresses the majority of your concerns and questions, aiding in a clearer understanding of our paper. Your feedback is valuable, so please don't hesitate to provide additional comments or ask further questions.

---

> > > > ### Author Response · Authors · 2023-11-22
> > > > **Sincerely looking forward to your feedback before deadline**
> > > >
> > > > Dear Reviewer 84Pi,
> > > >
> > > > Thank you very much again for your initial review of our paper. Given it is very close to the discussion deadline, and we are not sure whether you are satisfied with our answers and revised paper or whether there are still some concerns that have not been fully resolved, we would like to kindly remind you that we have answered all your questions and concerns mentioned in the weaknesses. Our latest revision provides a clearer picture of our work. We appreciate your time and invaluable feedback.
> > > >
> > > > Best,

---

### Author Response · Authors · 2023-11-14
**Global reply**

Dear all reviewers and ACs,

We would like to use the global response to consolidate the significant contributions of our paper. We offer a concise summary of our contributions and related works for the benefit of all reviewers and ACs. Additionally, the experimental results concerning our numerically stable gradient estimator in log space can be found in the latest revision's appendix.

| our contributions | previous literatures |
| - | - |
| motivate from the effectiveness of IS | [3] [5] [6] [7] |
| aim at learning $\theta$ | [1] [3] [4] [5] [6] [7] |
| no restrictions on the $q$ distribution families | [1] [2] [3] |
| directly minimizing forward $\chi^2$ divergence without surrogate | [2] [3] [5] [7] |
| motivate from the bias of IS in log space |  |
| numerically stable gradient estimator in log space |  |
| extensive experiments on cases where no explicit decomposition $p(\boldsymbol x,\boldsymbol z;\theta)=p(\boldsymbol x\vert\boldsymbol z;\theta)p(\boldsymbol z;\theta)$ |  |
| visualization for inferred latent and parameter $\theta$'s recovery |  |

* **Motivate from the bias of IS in log space**: All previous works have motivated $\chi^2$ from minimizing the effectiveness. We start by comparing the bias of the $\ln \hat p(\boldsymbol x;\theta,\phi)$ and $\widehat{\operatorname{ELBO}}(\boldsymbol x;\theta,\phi)$ to analyze why doing IS and the optimal way of doing IS. And the conclusion about minimizing forward $\chi^2$ divergence coincides with improving the effectiveness of the IS estimator (Fig. 1).
* **Numerically stable gradient estimator in log space**: Previous work already derived the gradient estimator for minimizing $\chi^2$ divergence in the original space but not in log space. This leads to the numerical instability issue and scaling to the high dimensionality issue. We argue that it is critical to estimate its gradient in log space to obtain a numerically stable and succinct form of the gradient estimator, especially for the score function estimator. Deriving the gradient estimator in the log space is mathematically non-trivial. We include the detailed derivation in the appendix. **Results in Appendix 5 and Fig. 7 in our latest revision implies that our score function gradient estimator is robust and scalable to a wide range of distribution families and latent space structures.**
* **Extensive experiments on cases where no explicit decomposition $p(\boldsymbol x, \boldsymbol z;\theta)=p(\boldsymbol x\vert\boldsymbol z;\theta)p(\boldsymbol z;\theta)$**: Most of the previous work only do experiments on generative model with explicit decomposition $p(\boldsymbol x,\boldsymbol z;\theta)=p(\boldsymbol x\vert\boldsymbol z;\theta)p(\boldsymbol z;\theta)$. However, when such an explicit decomposition does not exist, like the POGLM, and when the generative posterior distribution $p(\boldsymbol z|\boldsymbol x;\theta)$ and the approximating posterior distribution $q(\boldsymbol z|\boldsymbol x;\phi)$ are not Gaussian (which is usually assumed when using VI), ELBO cannot be reformulated as $\operatorname{ELBO}(\boldsymbol x;\theta,\phi) = \mathbb E_{q}[\ln p(\boldsymbol x|\boldsymbol z;\theta)] - \operatorname{KL}(q(\boldsymbol z|\boldsymbol x;\phi)\\|p(\boldsymbol z;\theta))$, and hence ELBO lost of a lot of advantages. Therefore, we do need a variety of graphical models to understand the performance of different methods.
* **Visualization for inferred latent and parameter $\theta$'s recovery**: Although theoretical materials show the superiority of VIS, practical visualization of the behavior of different methods is still necessary for us to get an intuition of how and why VIS performs better than others.

[1] Burda, Y., Grosse, R., & Salakhutdinov, R. (2015). Importance weighted autoencoders. arXiv preprint arXiv:1509.00519.
[2] Dieng, A. B., Tran, D., Ranganath, R., Paisley, J., & Blei, D. (2017). Variational Inference via $\chi $ Upper Bound Minimization. Advances in Neural Information Processing Systems, 30.
[3] Finke, A., & Thiery, A. H. (2019). On importance-weighted autoencoders. arXiv preprint arXiv:1907.10477.
[4] Jerfel, G., Wang, S., Wong-Fannjiang, C., Heller, K. A., Ma, Y., & Jordan, M. I. (2021, December). Variational refinement for importance sampling using the forward kullback-leibler divergence. In Uncertainty in Artificial Intelligence (pp. 1819-1829). PMLR.
[5] Domke, J., & Sheldon, D. R. (2018). Importance weighting and variational inference. Advances in neural information processing systems, 31.
[6] Su, X., & Chen, Y. (2021). Variational approximation for importance sampling. Computational Statistics, 36(3), 1901-1930.
[7] Akyildiz, Ö. D., & Míguez, J. (2021). Convergence rates for optimised adaptive importance samplers. Statistics and Computing, 31, 1-17.

---

### Author Response · Authors · 2023-11-21
**Hope to hear your feedbacks**

Dear reviewers,

According to all your valuable questions and suggestions, we have iterated several versions of our paper. There are a lot of changes in the main content and new results and figures in the Appendix in our latest revision. Hopefully, these new materials in our latest revision could solve most of your questions and concerns.

Since it is already very close to the discussion deadline, if you still have further questions about our work, please don't hesitate to ask us as soon as possible. We will try our best to answer them before the discussion deadline.

Thanks again for your time and valuable comments.

---

### Meta-Review · Area_Chair_oSo4 · 2023-12-10

**Metareview:**

This paper proposed an interesting variational importance sampling (VIS) algorithm to improve variational inference. Although there exist lots of work in the area, the proposed work appears to be novel and experimental results looks quite promising.

Strengths:
1. The proposed method appears to be novel with some good insights.
2. It could be potentially useful for general tasks using variational inference.

Weakness:
1. The writing and organizing need improvements.
2. Experimental results could be more comprehensive, right now, mainly on synthetic, MNIST as well as RGC. These are all small data sets. So potential of this method working in larger-scale data/models is not clear, at least empirically.

**Justification For Why Not Higher Score:**

The effect on more realistic data is not clear.

**Justification For Why Not Lower Score:**

The method appears to be novel and could have a broad impact on improving variational inference.

---

### Decision · Program_Chairs · 2024-01-16

Accept (spotlight)